



# A process-based model for quantifying the effects of canal blocking on water table and CO2 emissions in restored tropical peatlands

Iñaki Urzainki[1,2], Marjo Palviainen[3], Hannu Hökkä[4], Sebastian Persch[5], Jeffrey Chatellier[5], Ophelia Wang[5], Prasetya Mahardhitama[5], Rizaldy Yudhista[5], and Annamari (Ari) Laurén[2,3]

[1]Natural Resources Institute Finland (Luke), Latokartanonkaari 9, FI-00790 Helsinki, Finland
[2]School of Forest Sciences, Faculty of Science and Forestry, University of Eastern Finland, Joensuu Campus, P.O. Box 111, (Yliopistokatu 7), FI-80101 Joensuu, Finland
[3]University of Helsinki, Department of Forest Ecology, P.O. Box 27, FI-00014 Helsinki, Finland
[4]Natural Resources Institute Finland, Oulu, Paavo Havaksen tie 3, FI-90570 Oulu, Finland
[5]Forest Carbon PTE LTD, Singapore

**Correspondence:** Iñaki Urzainki (inaki.urzainqui@luke.fi)

**Abstract.** Drainage in tropical peatlands increases $CO_2$ emissions, the rate of subsidence, and the risk of forest fires, among other negative environmental impacts. These effects can be mitigated by raising the water table depth (WTD) using canal or ditch blocks. The performance of canal blocks in raising WTD is, however, poorly understood, because the WTD monitoring data is limited and spatially concentrated around canals and canal blocks. This raises the following question: how effective are canal blocks in raising the WTD over large areas? In this work we composed a process-based hydrological model to assess the rewetting performance of 168 canal blocks in a 22000 ha peatland area in Sumatra, Indonesia. We simulated daily WTD over one year using an existing canal block setup and compared it to the situation without blocks. The study was performed across two El-Niño Southern Oscillation (ENSO) scenarios, and four different peat hydraulic properties. Our simulations revealed that while canal blocks had a net positive impact on WTD rise, they lowered WTD in some areas, and the extent of their effect over one year was limited to a distance of about 600 m around the canals. We also show that canal blocks are most effective during dry periods and in peatlands with high hydraulic conductivity. Averaging over all modelled scenarios, blocks raised the annual mean WTD by only 0.9 cm. This value was 2.78 times larger in the dry year than in the wet year (1.39 cm versus 0.50 cm), and there was a 2.76 fold difference between the scenarios with the maximum and minimum hydraulic conductivity (1.50 cm versus 0.54 cm). Using a linear relationship between WTD and $CO_2$ emissions, we estimated that, averaging over peat hydraulic properties, canal blocks prevented the emission of 1.03 $Mgha^{-1}$ $CO_2$ in the dry year and 0.37 $Mgha^{-1}$ $CO_2$ in the wet year.

## 1 Introduction

Tropical peatlands contain approximately one sixth of the global soil carbon pool (Page et al., 2022, 2011; Xu et al., 2018). In the recent decades, extensive tropical peatland areas have been converted to agricultural and plantation forest production (Wijedasa et al., 2018). This land use change has often been driven by drainage, which involves excavating canals or ditches in the peat. Canals help to remove water from the naturally waterlogged peat, enhancing site productivity, and opening pathways





for wood and crop transportation (Dohong et al., 2017). However, the same mechanisms that make the drainage-based bio-production economically valuable have severe environmental consequences. Drainage increases $CO_2$ emissions (Novita et al., 2021; Jauhiainen et al., 2012; Ishikura et al., 2018; Carlson et al., 2015), the rate of peat subsidence (Evans et al., 2022, 2019; Hooijer et al., 2012; Sinclair et al., 2020; Hoyt et al., 2020), fire risk (Miettinen et al., 2017; Kiely et al., 2021), nutrient release (Laurén et al., 2021a, b) and export to water courses (Nieminen et al., 2017), and decreases the peat substrate quality (Könönen et al., 2018).

Drainage lowers the peatland water table depth (WTD, meters, negative downward), which onsets mechanisms that are behind the environmental drawbacks. The lower WTD increases the oxygen supply that soil microorganisms need for aerobic decomposition of organic matter (Page et al., 2022). As a result of the decomposition process, $CO_2$ is emitted, peat subsides, and nutrients are released. Therefore, raising the WTD has been the focus of many restoration practices. Canal blocks or dams raise the canal water level (CWL), increase the residence time of water in the peatland, and raise the WTD in the peat (Dohong et al., 2018).

Despite the widespread use of canal blocks for peatland restoration, there exists little evidence for their effectiveness, especially in large areas. Most existing studies monitor WTD before and after block installation using dipwells. Due to practical restrictions, dipwells are usually installed close to the canals, which is the area where WTD rise due to canal blocks is expected to be largest (Sutikno et al., 2020; Kasih et al., 2016; Ritzema et al., 2014). As a result, a naive extrapolation of the observed block-induced WTD response to larger scales will likely result in overestimating their effectiveness. Moreover, since WTD depends on variable meteorological factors and complex hydrological processes, the difference between WTD before and after building the blocks cannot be directly attributed to their presence. In their review about tropical peatland restoration practices, Dohong et al. (2018) concluded that while nearly all canal blocking studies have reported that the WTD rose after the dams were placed, "our current knowledge and skills are arguably inadequate for the large and landscape scale peatland restoration in Indonesia".

Process based models offer a different, complementary approach to analyze WTD response to blocks. If implemented correctly, the models can account for the complex, interconnected factors affecting the canal block WTD response in large areas, which include peat topography, canal topology and block location, peat hydraulic properties, and rainfall patterns. They also enable a direct comparison of WTD between different blocking setups. Process based models have been applied to simulate WTD in tropical peatlands in multiple studies (Wösten et al., 2006; Cobb et al., 2017; Baird et al., 2017; Urzainki et al., 2020). Only few of those have dealt with the question of block performance (Jaenicke et al., 2010; Ishii et al., 2016; Putra et al., 2022). The studies by Ishii et al. (2016) and Jaenicke et al. (2010) did not consider different peat hydraulic properties or weather scenarios, therefore limiting the generalizability of their results. Putra et al. (2022), on the other hand, presented a good experimental setup to analyze block efficiency, but their simulations were confined to an area of 20 ha. Notwithstanding the usefulness of their approach to plan small-scale mitigation strategies and to understand restored peatland WTD dynamics, such small scales are insufficient for assessing the rewetting abilities of blocks over regional scales.

The aim of the present study is to assess the effectiveness of canal blocking restoration practices for a large tropical peatland area in Sumatra, Indonesia, using a process-based hydrological model. We seek to understand the scale of the block impact





under different weather conditions and peat hydraulic properties. To meet that challenge, we constructed a new hydrological model that combines the diffusive wave approximation of the open channel flow equations with the groundwater flow equation that solves the WTD throughout the peatland area. The model for the CWL is sensitive to the presence of canal blocks, and

the spatially-explicit WTD was used to compare the blocked and non-blocked scenarios. The results were further evaluated to assess the impact of canal blocking on $CO_2$ emissions from the tropical peat area.

## 2 Materials and Methods

### 2.1 Study area

The 22000 ha study area is part of an ecosystem restoration concession, the Sumatra Merang Peatland Project (SMPP), which

is located in within the largest peat swamp dome in South Sumatra—the 140000 ha Merang-Kepayang peat dome. The area is an ecologically significant wetland close to Berbak Sembilang National Park, and as many other swamp forests in Southeast Asia, it has been degraded by logging of the primary forest and by the construction of drainage canals. After more than a decade of widespread illegal logging, the SMPP rehabilitation project began in 2017 with an initial installation of 87 temporary box dams, followed by 203 permanent peat compaction dams that were constructed between 2019 and 2021. The SMPP area

remains uninhabited, and pioneering native forest species are the main vegetation cover, with only 200 ha of original peat swamp forest habitat remaining.

Our study site, a large subset of the SMPP area, contains 219 km of canals and 168 dams (Figure 1). The locations of the peat compaction dams were based on elevation difference by distance (Jaenicke et al., 2010). The peat depth averages at about 5 m. There are five patrol posts inside our area, each consisting of a weather station and six daily-measured dipwells along a

200 m transect perpendicular to the nearest canal. There are 111 additional dipwells measured manually at an approximately monthly frequency.





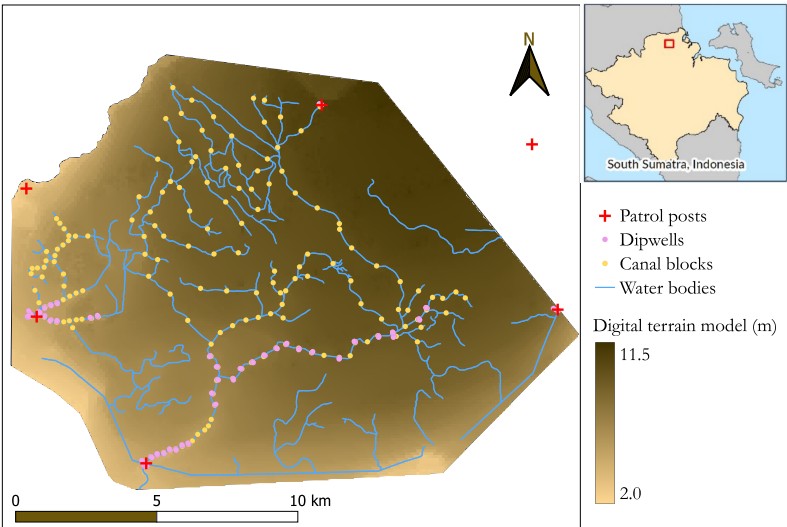

**Figure 1.** Study area. Digital terrain model (brown gradient), water bodies (blue lines), dipwells (pink dots), canal blocks (yellow dots) and patrol posts (red crosses). Each patrol posts consist of a weather station and six daily measured dipwells along a 200 m transect. The rest of the dipwells were measured manually with monthly frequency. Native forest species are the main vegetation cover.

## 2.2 Modelling

We constructed a hydrological model that produces daily maps of WTD. The model consists of a canal network module (CNM) and a peat hydrological module (PHM). At each timestep, these modules work in an alternate fashion to update the next day's

canal water level (CWL, meters, negative downwards) and WTD across the study area. First, the CNM computes and updates the CWL using the amount of water expected to have flowed in the peatland-canal interface, which was computed by the PHM in the previous timestep. Then, the PHM computes and updates the WTD using the newly computed CWL. The PHM allows for bidirectional water flow between the canals and the peatland, but it only updates the state of the WTD, not the CWL.

The CWL and the WTD are essentially the same quantity: water height above a common reference datum; therefore, they

are described in this text with the same symbol, $h$. The context is hopefully clear enough for the reader to discriminate between the two. In the following, each module is described in more detail.

### 2.2.1 Canal network module (CNM)

The CNM solves $h$ in the canal network using a diffusive wave approximation of the open channel flow equations (Szymkiewicz, 2010). This approximation requires less computational resources than a solution of the full equations, making it particularly

suitable for catchment-scale peatland areas with complex canal structures. Additionally, it is able to describe the propagation of the water flow both in the upstream and downstream directions and thus it can represent the upstream influence of dams, a key feature for our intended application. The diffusive wave approximation is derived from the open channel flow equations by





neglecting the two inertial terms in the momentum equation, which results in a gradient of $h$ that depends only on the friction slope (Novák, 2010). Here we use a formulation of the open channel flow equations given by the water surface elevation from the reference datum, $h$ [m], and the discharge, $Q$ [m³s⁻¹], with the friction slope described by Manning's equation (Cunge et al., 1980),

$$\frac{\partial h}{\partial t} = -\frac{1}{B}\frac{\partial Q}{\partial x} + \frac{q}{B} \tag{1}$$

$$\frac{\partial h}{\partial x} = -\frac{n^2 Q|Q|}{A^2 R^{4/3}}. \tag{2}$$

Here $B$ is the channel width [m], $q$ is the lateral inflow per unit length [m³m⁻¹s⁻¹], $A$ is the cross-sectional flow area [m²], $n$ is the Manning friction coefficient [m⁻¹ᐟ³s] and $R$ is the hydraulic radius [m]. Our model used a simple rectangular channel of height $z$, i.e., $A = B(h - p + z)$, where $p$ [m] is the local peat surface elevation above the reference datum. A graphical representation of the variables is shown in Figure 2, and Table 1 contains the parameter values.

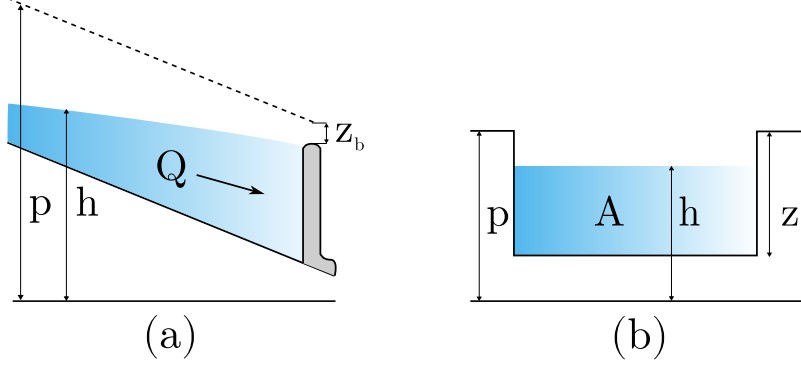

**Figure 2.** Schematic representation of the relevant variables in the open channel flow equations. **(a)** Side view and **(b)** channel cross-section. The grey structure represents a dam.

The mass conservation equation, Eq.(1), may be combined with the momentum equation, Eq.(2), to eliminate one of the dependent variables, $h$ or $Q$. Usually, $h$ is eliminated to get an advection-diffusion partial differential equation (PDE) for $Q$ (Novák, 2010; Szymkiewicz, 2010). However, here we are interested in the water level in the canal network, and thus instead eliminate $Q$ to get a PDE for $h$. This transformation and the resulting conservative numerical schemes are expressed in detail in Appendix A.

We constructed the Manning friction coefficient $n$ according to these assumptions:

– The friction increases as the CWL approaches the canal bed. This is due to the resistance to water flow introduced by vegetation growing in the canals and the canal bed surface roughness.

– When the CWL is below the canal bed, water may still flow through the underlying peat. The friction coefficient in this zone must be several orders of magnitude higher because it is effectively describing water flow in a porous medium.





**Table 1.** Fixed parameter values across all modelled scenarios.

| Parameter | Value | Unit | Equation | Description |
|---|---|---|---|---|
| $B$ | 1.5 | m | (1) | canal width |
| $z$ | 1.5 | m | (3) | canal height. Distance to canal bed measured from the local peat surface |
| $z_b$ | 0 | m | (4) | Distance to the block head, measured from the local peat surface |
| $n_t$ | 100 | $\mathrm{m}^{-1/3}\mathrm{s}$ | (3) | Maximum value of $n$. |
| $n_1$ | 5 | - | (3) | Parameter of the Manning friction coefficient |
| $n_2$ | 1 | - | (3) | Parameter of the Manning friction coefficient |
| $K_b$ | 2000 | $\mathrm{m}^{3/2}\mathrm{s}^{-1}$ | (4) | Block discharge coefficient |
| $s_1$ | 0.6 | - | (8) | Parameter of the specific yield function |
| $s_2$ | 0.5 | - | (8) | Parameter of the specific yield function |

  – The friction increases with decreasing CWL in a nonlinear fashion.

The Manning friction coefficient was described as follows:

$$n(h) = \begin{cases} n_t e^{-n_1(h-p+z)^{n_2}} & h > p - z \\ n_t & h \leq p - z \end{cases}. \tag{3}$$

Here $n_t$ is a threshold value of $n$ [$\mathrm{m}^{-1/3}\mathrm{s}$], and $n_1$ and $n_2$ are parameters. When the CWL drops below the canal bed elevation $h_b$, the Manning friction coefficient is equal to its maximum, $n_t$. For heads above the canal bed, $n$ decreases exponentially with a shape dictated by $n_1$ and $n_2$. In the absence of better information sources the values for these parameters were chosen by trial and error (see Table 1). With the choice of parameters presented in Table 1, $n = 0.055 \,\mathrm{m}^{-1/3}\mathrm{s}$ when the canal is full of

water, which is in the rage described in the literature (Szymkiewicz, 2010), and $n = 100 \,\mathrm{m}^{-1/3}\mathrm{s}$ for flow below the canal bed.

The water discharge through a canal block, $Q_b$ [$\mathrm{m}^3\mathrm{s}^{-1}$], was modelled using the following relationship (Szymkiewicz, 2010),

$$Q_b = K_b \max\left(0, h - p + z_b\right)^{1.5}, \tag{4}$$

where $K_b$ [$\mathrm{m}^{3/2}\mathrm{s}^{-1}$] is a coefficient regulating the rate of water flow through the block, and $p - z_b$ [m] is the elevation of the

block head above the reference datum (Figure 2. With this choice, the dam completely blocks water when the CWL is below the block head level.

The challenge of solving the open channel flow equations in a large network of interconnected canals was met with a novel discretization of the equations. As usual, each canal reach was discretized as a one dimensional grid with a fixed spacing between nodes of $\Delta x = 50\mathrm{m}$. The novelty introduced by our method concerns the canal junctions. We exploited the basic

properties of conservation equations that fully specify the governing equations at every node in the computational domain.





This is different from the usual practice, in which external mass and energy conservation equations need to be added manually to the system of equations in order to describe water flow at canal junctions (Cunge et al., 1980). As a result, the computational domain in our method is, by design, analogous to the canal network topology, which simplifies the implementation. Further details on the discretization method are given in Appendix A.

In our implementation of the model, disconnected components of the canal network were solved independently, in parallel processes. The accelerated Newton-Raphson method introduced by Liu and Hodges (2014) was used to solve the resulting linear system of equations. No-flow Neumann boundary conditions were set at all boundary nodes.

### 2.2.2   Peat hydrological module (PHM)

This module uses the output from the CNM to compute the daily WTD in the peat. Fundamentally, the approach is very similar
to the peat hydrological module presented previously in Urzainki et al. (2020) and many others before that (see, e.g., Cobb et al. (2017); Morris et al. (2012); Putra et al. (2022)). However, the implementation details differ slightly. We solve the two dimensional groundwater flow equation, which is suitable for domains much wider than they are thick (Connorton, 1985; Bear and Cheng, 2010). In this work we formulated it in terms of the water storage, $\Theta$ [m],

$$\frac{\partial \Theta}{\partial t} = \nabla \left( D(\Theta) \nabla \Theta \right) + P - ET, \tag{5}$$

where $P - ET$ [md$^{-1}$], the difference between precipitation and evapotranspiration, is the net water input to the system, and $D$ is the diffusivity [m$^2$d$^{-1}$], which is defined as

$$D(\Theta) = \frac{T(\Theta)}{S_y(\Theta)}, \tag{6}$$

where $T$ [m$^2$d$^{-1}$] is the transmissivity and $S_y$ is the specific yield.

Equation (5) describes water flow through a porous medium. When water ponds above the peat surface, the medium in which
water moves is no longer porous, and the physical description of the dynamics of water changes. Our model explicitly separates the two domains (below and above ground) by using piecewise-defined peat hydrological properties. We avoided any potential numerical problems caused by the discontinuities of the peat hydraulic properties at the domain threshold by imposing the continuity and differentiability of $D(\Theta)$ at this point. For more details, see Appendix B.

Both the specific yield and the transmissivity are known to vary depending on WTD. This is especially true of the trans-
missivity, which may vary in several orders of magnitude in just a few centimetres (Cobb et al., 2017). Following the results of Cobb and Harvey (2019), our model describes the nonlinear variation of $S_y$ and $T$ in the vertical profile with exponential functions. While the formulation in terms of $\Theta$ was used for the numerical implementation of the groundwater flow equation, Eq.(5), it is easier to interpret hydrological variables in terms of

$$\zeta = h - p, \tag{7}$$



the WTD [m, negative downwards]. We leave the details of the formulation of the peat physical properties in terms of $\Theta$ to Appendix B. The specific yield was parameterized as:

$$S_y(\zeta) = \begin{cases} 1 & \zeta > 0 \\ s_1 e^{s_2 \zeta} & \zeta \leq 0 \end{cases}, \tag{8}$$

where $s_1$ and $s_2$ are parameters (see Table 2). When the WTD is above ground, the theoretical value of the specific yield is $S_y = 1$, because a unit increase in water volume per unit area increases the WTD by the same amount.

The transmissivity was parameterized as:

$$T(\zeta) = \begin{cases} \alpha\zeta + \beta & \zeta > 0 \\ t_1(e^{t_2 \zeta} - e^{-t_2 d}) & \zeta \leq 0 \end{cases}, \tag{9}$$

where $d$ is the local peat depth [m, negative downwards] and $t_1$ and $t_2$ are parameters (see Table 2).

Below ground, the transmissivity increases exponentially with WTD from a value of $T(\zeta = -d) = 0$ when the WTD is at the bottom of the peat column. Above ground, on the contrary, we opted for a constant conductivity, the derivative of transmissivity

($K = \frac{\partial T}{\partial \zeta}$), which results in a linear $T$.

The parameters $\alpha$ and $\beta$ were chosen so that $D(\Theta)$ was continuous and differentiable at $\Theta(\zeta = 0)$. As shown in Appendix B, these values are:

$$\begin{cases} \alpha & = \frac{t_1}{s_1^2}\left(t_2 - s_2 + \frac{s_2}{e^{t_2 d}}\right) \\ \beta & = \frac{t_1}{s_1}\left(1 - e^{-t_2 d}\right) \end{cases}. \tag{10}$$

Equation (5) was solved using an implicit finite volume solver in an unstructured mesh generated from the study area maps.

Our code relied on open source software (Guyer et al., 2009; Geuzaine and Remachle, 2009). Fixed head Dirichlet boundary conditions with value $\zeta = -0.2m$ were applied at the domain boundaries.

### 2.2.3  Module interaction

Each timestep, the CNM and the PHM are executed in an alternate fashion. The two modules operate in different computational domains, and thus water flow between the canal network and the peat matrix has to be specified externally. On the one hand,

the CNM receives information about the peat WTD through the lateral water inflow, $q$ (see Eq.(1)). On the other, the CWL computed in the same timestep is used to populate the PHM mesh cells corresponding to the canal network, thus informing the PHM about the latest CWL status.

In the solution of the PHM, so as not to compute canal flow twice, water flow between any two adjacent mesh cells that corresponded to the canal network was completely restricted by setting $D = 0$ in the cell faces. In contrast, water flow between

canal and peat cells was allowed. The execution of the PHM did not directly modify the CWL; instead, the head difference at canal cells before and after the execution of the PHM was used to compute the lateral water inflow $q$ for each timestep. Thus,



$q$ acts as a sink/source term which captures how much water is expected to enter or leave each node of the canal network in the next timestep. Whenever the CWL rose above ground, the volume of ponding water was distributed instantaneously throughout the cell area in the PHM step.

An hourly timestep was used in each internal iteration. although smaller timesteps were adopted if any of the modules had not converged to a specified accuracy.

### 2.2.4    Input requirements

The model runs with easily available GIS data: maps of surface elevation and peat depth, as well as a vector file that specifies the topology of the canal network. The digital elevation model and peat depth maps has a resolution of 50 m × 50 m, and

they were derived from high-resolution light detection and ranging (LiDAR) data collected by Deltares following the methods described in Vernimmen et al. (2020). The peat depth model was derived from a geographically weighed regression and spatial interpolation of a peat thickness field inventory. Additionally, daily weather information is required for the solution of Eq.(5). Each cell of the finite element mesh receives a daily source term input given by the difference between precipitation and evapotranspiration, $P - ET$. The weather data used in this study differed between model scenarios, the following section for

more details.

### 2.3    Modeled scenarios

We simulated the WTD over one year for 16 different scenarios. Each simulation started from the same initial WTD. We ran the model with two different weather data (dry and wet years); two different blocks states (blocked or without any blocks); and four different peat hydraulic properties functions, $S_y$ and $T$. The 16 different scenarios arise from a combination of all of the

above factors ($2 \times 2 \times 4 = 16$).

Apart from those, we also conducted a simple reality check for the model, in which we compared the model results with WTD sensor data. The reality check was computed using locally measured weather station data for a single set of values of the peat hydraulic properties. This was only meant as an informal check of the plausibility of our model, and was not a part of the main results of this work.

### 2.3.1    Weather scenarios

The two major components of the water balance in tropical peatlands, precipitation and evapotranspiration, enter the PHM as a net water sink/source term in the PDE, Eq.(5). Precipitation data was collected from the Sulthan Thaha Airport weather station, the nearest BMKG (Indonesian Meteorology, Climatology and Geophysics Agency) weather station available. One dry year (1997) and one wet year (2013) were selected, from more than 30 years of data, according to total annual rainfall. Net

annual water input between the two scenarios differed in 1255 mm: Total rainfall in the dry and wet years were 1293.5 mm and 2584.0 mm, respectively. Around day 150 in the dry scenario a prolonged dry period began, which lasted almost until the end of the year. The wet year had intense and prolonged rainfalls, even during the dry period. The resulting net water sources





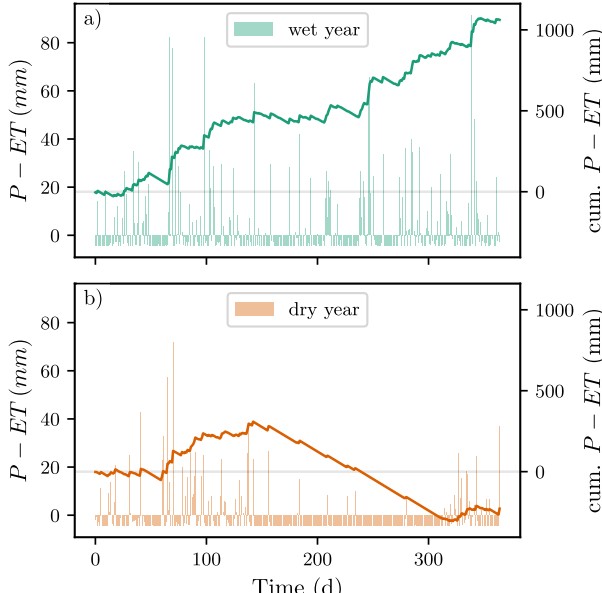

**Figure 3.** Net water source/sink term (precipitation minus evapotranspiration) in the wet **(a)** and dry **(b)** modelled scenarios. Vertical bars show daily net water source, and the solid line shows the net cumulative sink/source of water. Note that only the baseline evapotranspiration of $ET = 4.17$ mmd$^{-1}$ is shown here; the additional contribution due to the pan evaporation term was dependent on WTD and differed across modeled scenarios.

for the two scenarios are shown in Figure 3. The two selected years reflect extremes of the large inter-annual and seasonal variability that are common in the tropics due to the El Niño–Southern Oscillation (ENSO).

The evapotranspiration was modelled as a constant daily value plus a pan evaporation term that only became active when the WTD was close to the peat surface. The variation of evapotranspiration in tropical peatlands is considerably smaller than that of rainfall. This applies both to inter-annual variations in total annual evapotranspiration as well as to daily and seasonal variations within a year (Hirano et al., 2015; Wati et al., 2018). Evapotranspiration enters our model as a quantity relative to precipitation, which justifies our choice of a constant baseline. This value was $ET = 4.17$ mmd$^{-1}$, the mean of 7 years of

measurements across tropical peatlands in different disturbance states (Hirano et al., 2015). Wati et al. (2018) found that pan evaporation (evaporation from ponding water) in Java and Bali could be as high as $7$ mmd$^{-1}$. Based on that, we added a pan evaporation term to the constant baseline. This pan evaporation term increased linearly from $0$ mmd$^{-1}$ when the WTD was at $-10$ cm, to $3$ mmd$^{-1}$ when the WTD was at $+10$ cm. The contribution of the added pan evaporation term for higher water levels was cut off at $3$ mmd$^{-1}$.





**Table 2.** Parameters of the peat hydraulic properties, Eqs.(8) and (9). The peat hydraulic properties resulting from these parameters are shown in Figure 4. These four sets of parameters were used in different modelled scenarios.

| Name | $s_1$ | $s_2$ | $t_1$ | $t_2$ |
|------|-------|-------|-------|-------|
| 1 | 0.6 | 0.5 | 50 | 2.5 |
| 2 | 0.6 | 0.5 | 50 | 7.5 |
| 3 | 0.6 | 0.5 | 500 | 2.5 |
| 4 | 0.6 | 0.5 | 500 | 7.5 |

### 2.3.2 Block configurations

Two different dam setups were modelled. One setup, which will be referred to as 'blocked configuration' or simply 'blocked', consisted of all the 168 blocks present in the study area (see Figure 1). The other setup, called 'not blocked', did not contain any blocks.

### 2.3.3 Peat hydraulic properties

Our description of the peat hydraulic properties was based on the findings presented in Cobb et al. (2017), Hooijer (2005) and Baird et al. (2017). The values reported for the transmissivity and its derivative, the conductivity, $K$, vary ostensibly both between sites and in the vertical soil profile within the same site. In their measurements in several Panamanian peatlands, Baird et al. (2017) found that hydraulic conductivities at a depth of around $0.5\,\mathrm{m}$ in the peat profile were in the range $K = 7.5$–$471.9$ $\mathrm{md}^{-1}$. In an Indonesian peatland, Cobb et al. (2017) found $K = 1300\,\mathrm{md}^{-1}$ at the surface and $K = 5.2\,\mathrm{md}^{-1}$ at 30 cm below the surface. The specific yield, on the other hand, varies less in the cited studies. All values are in the range $S_y = 0.29$–$0.68$, deeper layers of the peat profile having lower values. To capture some of this range and the rapid vertical change in the soil profile, we chose to use a single specific yield curve and four different transmissivity curves, which were modelled using Eqs.(8) and (9). The resulting peat hydraulic properties are shown in Figure 4, and the sets of parameters used to generate them are listed in Table 2.

### 2.3.4 Reality check

We performed a reality check of our model by comparing the simulated WTD with the available measured field data. The variation in peatland topography at smaller scales than the resolution limit of our data ($50\,\mathrm{m} \times 50\,\mathrm{m}$) prevented any meaningful one-to-one quantitative comparison between the modelled and the measured WTD. This small scale variation in peat elevation is known to be of tens of centimeters (Lampela et al., 2016), which is comparable to daily and annual WTD ranges.

The field WTD was measured from the dipwells (see Figure 1). We modelled the WTD for one year for all peat hydraulic properties using the blocked setup.





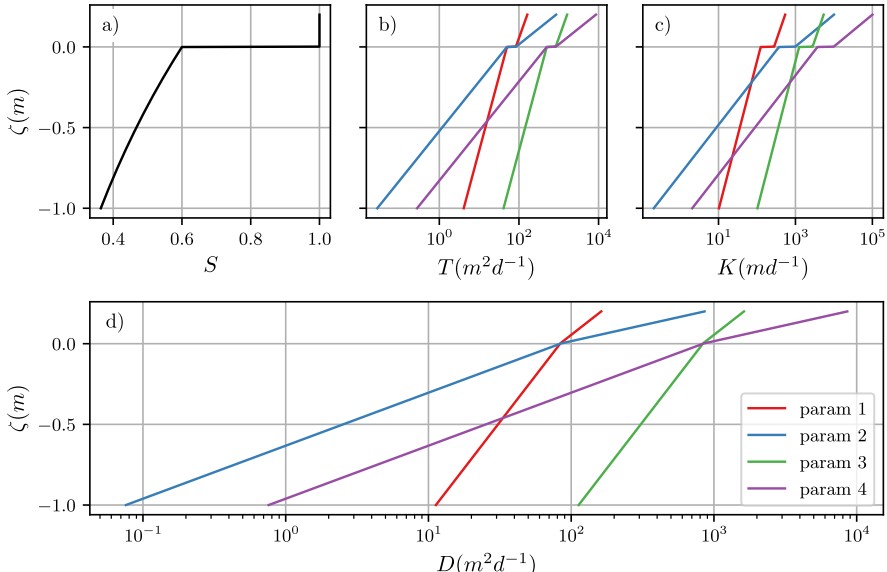

**Figure 4.** The four modelled sets of peat hydraulic properties arising from the parameters in Table 2. **(a)** Specific yield, Eq.(8), common to all modelled scenarios. **(b)** Transmissivity, Eq.(9). **(c)** Conductivity, the derivative of transmissivity, $K = \frac{\partial T}{\partial \zeta}$. **(d)** Diffusivity, Eq.(6). **(b)**, **(c)** and **(d)** are shown in a logarithmic scale. The different parameter sets are color coded, and this code is used in the rest of the text: Red, blue, green and purple correspond to parameters 1,2,3 and 4, respectively.

Precipitation and evapotranspiration were determined using the data collected at the six patrol posts' weather stations (see Figure 1). The weather data was gathered during the year 2020, and it included recordings of daily precipitation, temperature, windspeed, atmospheric pressure and relative humidity. The daily precipitation was adopted directly from the weather station

measurements. The evaporation was modelled using the standard Penman-Montheith equation (Allen et al., 1998), fitting the two free parameters of the model to the annual net radiation and evapotranspiration reported in (Hirano et al., 2015). Each cell in the PHM domain received a spatially interpolated daily $P - ET$ value that was based on its distance to the weather stations. We chose to present here the peat hydraulic property set that best fitted the below ground range and the dynamics of measured WTD.

**2.3.5 Initial condition**

The initial WTD was the same in all modelled scenarios, including the reality check, and it was derived as follows. Starting from total water saturation, $\zeta = 0$ m everywhere in the study area, we let the model evolve with no precipitation and a high evapotranspiration, $ET = 7.5$ mmd$^{-1}$. The simulations were run with blocks and with the set of peat hydraulic properties number 4 (see Table 2). The model was run for 50 days, recording the resulting WTD rasters at the end of each day. We then

compared those WTD rasters with the dipwell measurements of January 22nd, 2020, the first available sensor measurements of the year. The modelled WTD raster that best agreed with those sensor measurements was selected as the initial condition for





all scenarios. Compared to other possible choices for the initial state of WTD, such as a constant WTD throughout the area, this initial condition captures the natural curvature of the WTD.

### 2.4 Notation

We will use $\langle \zeta \rangle$ to indicate the spatially averaged WTD, and $\bar{\zeta}$ to indicate the temporal average. Additionally, the quantity

$$\Delta\zeta = \zeta_{blocks} - \zeta_{no-blocks} \tag{11}$$

will be used to indicate the WTD difference between two modelled scenarios that only differ by the blocking condition.

### 2.5 Translation to $CO_2$ emissions

$CO_2$ emissions in the study area were modelled as a linearly increasing function of WTD,

$$m_{CO_2}(\zeta) = -a\zeta + b, \tag{12}$$

where the negative sign implies that the emissions increase with deeper WTD (note that $\zeta$ is negative below ground). Equation (12) was used for below ground WTD. The $CO_2$ emissions resulting from above ground WTDs were set equal to the emissions at the surface, i.e., $m_{CO_2}(0) = b$. In this work, we used the values from Jauhiainen et al. (2012), $a = 74.11$ Mgha$^{-1}$m$^{-1}$yr$^{-1}$ and $b = 29.34$ Mgha$^{-1}$yr$^{-1}$. These values were obtained for an Acacia plantation, which may not give the most accurate
estimation of the emissions from a rehabilitating natural peat swamp forest. Therefore the reader is encouraged to treat the $CO_2$ emission results as a rough estimation of their magnitude.

Following the notation introduced in Eq.(11), we will denote the difference in $CO_2$ emissions due to the block influence by $\Delta m_{CO_2}$.

## 3 Results

### 3.1 Reality check

The comparison between the modelled and the measured WTD shows similarity in the range and the dynamics of WTD, as shown in Figure 5 (a). The deepest WTD in both measured and modelled scenarios was about $-1.5$ m. During most of the year WTD was above $-0.4$ m in the majority of the measuring locations. The daily fluctuations produced by variations in precipitation and evapotranspiration were comparable in their magnitude. The water rise and recession slopes were similar as
well. As a result, the distribution of annually averaged modelled WTD was very similar to the measured one, as can be seen in Figure 5 (b).





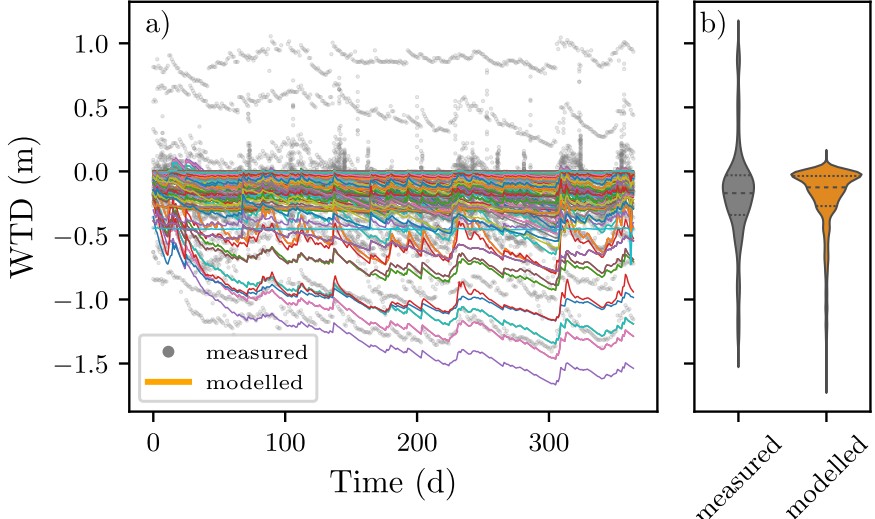

**Figure 5.** Outcome of the reality check. **a)** Measured WTD at the 141 dipwells (dots), and modelled WTD at the same locations (coloured lines). **b)** Kernel density estimation of measured and modelled WTD in a). The dotted lines indicate the quartiles of the distribution.

## 3.2 Block impact on WTD

Blocks led to a net rise of the WTD in all modelled scenarios. This is shown qualitatively in Figure 6, which aggregates $\Delta\bar{\zeta}$ from all modelled scenarios into a single raster. The existence of more areas with a positive $\Delta\bar{\zeta}$ (colored in blue in Figure 6) means that, overall, blocks produced a net WTD rise across all the modelled scenarios. Averaging over all modelling scenarios, the net average WTD rise was $9.4\,\mathrm{mm}$. Despite the overall WTD rise, in some areas blocks had the effect of lowering WTD compared to the non-blocked scenario (colored in red in Figure 6). It is also remarkable that the WTD in most of the peatland area far enough from canals was practically unaffected by the presence or absence of blocks.

The spatially averaged block-induced WTD rise, $\langle\Delta\zeta\rangle$, differed significantly across different hydraulic property values and weather scenarios, as shown in Figure 7. However, $\langle\zeta\rangle$ was always higher with the blocks than without the blocks (positive $\langle\Delta\zeta\rangle$) for all modelled scenarios. In other words, the overall rewetting impact of the blocks was positive at all times during the simulated period, regardless of weather conditions or peat hydraulic properties.

Block impact on WTD was greater in dry periods, when external water input to the system was scarce. Differences between blocked and non-blocked WTD increased with prolonged dry conditions and decreased with rainfall events. The extreme drought starting around day $150$ in the dry weather scenario provides a good example of this effect. As the dry period progressed, the gap between blocked and non-blocked WTD kept expanding for all peat hydraulic properties (Figure 7 (b) and (d)). Conversely, rainfall events reduced the difference between the WTD in the blocked and non-blocked scenarios (Figure 7 (b) and (c)). As a result, the cumulative block-induced WTD rise averaged over peat hydraulic properties was $2.78$ times larger in dry conditions than in wet conditions ($1.39\,\mathrm{cm}$ versus $0.5\,\mathrm{cm}$).

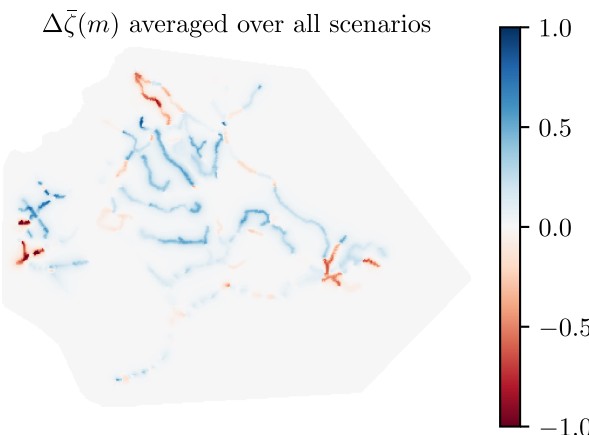

**Figure 6.** Temporal average of the water table depth (WTD) difference between the blocked and non-blocked scenarios, $\Delta\bar{\zeta}$, averaged over all modelled scenarios. Blue and red colors represent the areas in which blocks raised and lowered the WTD, respectively. The grey color present in most of the study area indicates a negligible effect of the blocks on WTD far away from canals, i.e., $\Delta\bar{\zeta} \approx 0$. The block positions are shown in Figure 1.

Peat hydraulic properties also played an important role in the degree to which blocks were able to raise WTD. Specifically, a larger hydraulic conductivity (or transmissivity) led to greater differences between the blocked and unblocked scenarios, in both weather conditions (Figure 7 (c) and (d)). Averaged over the two weather conditions, the $\langle\Delta\bar{\zeta}\rangle$ obtained with parameter set 3, was 2.76 times greater than the one obtained with parameter set 2 (1.5 cm versus 0.54 cm).

There was remarkable variation in the spatial extent of block influence among the different modelling scenarios. As an
example, we show two snapshots of $\Delta\zeta$ for the same peat hydraulic properties, parameter set 3, in wet and dry years in Figure 8 (b) and (c). These figures qualitatively show that the WTD difference due to the blocks at the end of the wet year was small and concentrated around canals. In contrast, the differences at the end of the dry year were both more pronounced and extended further spatially. Note that both the positive and the negative block impacts were larger at the end of the dry year, i.e., $\Delta\zeta$ was larger in absolute value. The dependence of $\Delta\bar{\zeta}$ on the distance to the nearest block, shown in Figure 8 a), provides a more
quantitative description of the spatial extent of block impact. The effect of blocks on WTD for all modelled scenarios was relevant until about 600 m from the nearest dam, and it markedly decreased from there. The mean annual impact of the blocks on WTD more than one kilometre away was negligible (not shown in Figure 8 (a) for clarity). This was true across all weather conditions and peat hydraulic properties, albeit with small variations between modelled scenarios. Drier conditions and higher hydraulic conductivities had the effect of increasing the spatial extent of the influence of blocks.

**3.3 Potential to decrease CO$_2$ emissions**

The net block-induced WTD rise in all modelled scenarios led to an overall decrease of CO$_2$ emissions, Figure 9. In the worst performing rewetting scenario, with wet conditions and a peatland with the lowest studied hydraulic conductivity (parameter





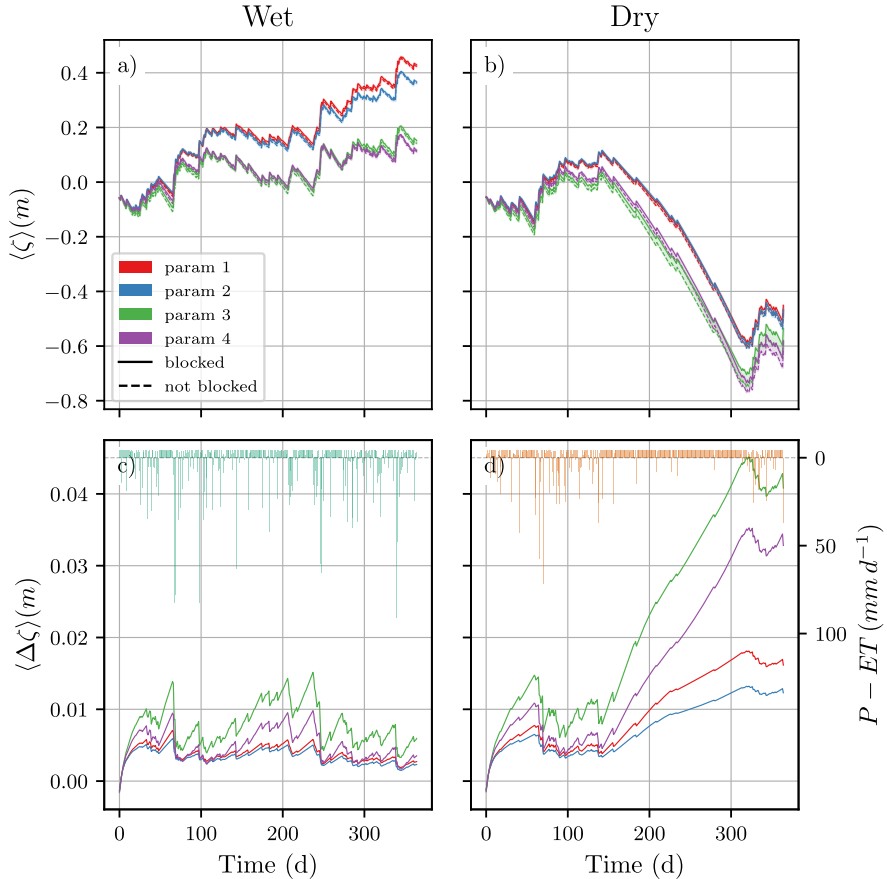

**Figure 7.** Spatially averaged WTD differences between the blocked and non-blocked scenarios for all the modelled scenarios. Spatially averaged WTD for wet **a)** and **b)** dry weather conditions. The solid lines correspond to the blocked scenario, and the dashed lines to the non-blocked. The shaded area between solid and dashed lines of the same color corresponds to the WTD difference between the two blocking scenarios, $\langle \Delta\zeta \rangle$, which is explicitly shown in **c)** and **d)**. Vertical bars in **c)** and **d)** show the daily net water input $P - ET$ [md$^{-1}$].

set 2), the non-blocked setup emitted an annual total of $0.28 \ \mathrm{Mgha}^{-1}$ more than the blocked one. In contrast, in the best-performing scenario (dry conditions, parameter set 3) the block-induced WTD rise was translated into $1.65 \ \mathrm{Mgha}^{-1}\mathrm{y}^{-1} \ CO_2$ emission reduction. Averaging over all peat hydraulic properties, the emission of $0.37 \ \mathrm{Mgha}^{-1}$ and $1.03 \ \mathrm{Mgha}^{-1} \ CO_2$ was prevented in the whole year for dry and wet years, respectively.






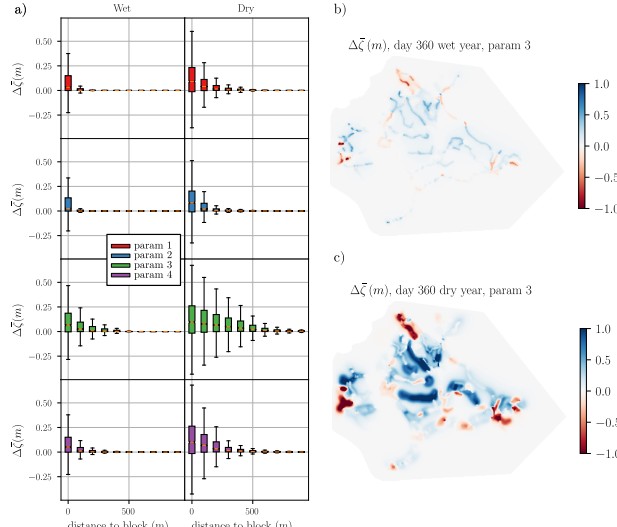

**Figure 8.** Spatial extent of the impact of blocks on WTD. **a)** Temporal average of $\Delta\zeta$ plotted against the distance to the nearest block, categorized in 100 m classes. The box plot extends from the first to the third quartile, with an orange line at the median. The whiskers extend until 1.5 times the inter-quartile range. **b)** and **c** show a snapshot of $\Delta\zeta$ at day 360 of the wet **b)** and dry **c)** years for the set of peat hydraulic properties number 3.

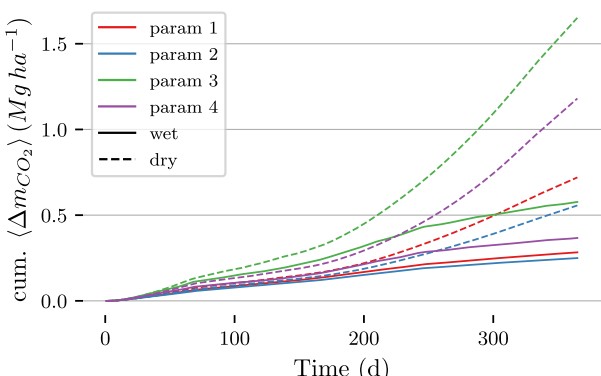

**Figure 9.** Cumulative change in $CO_2$ emissions due to the blocks, $\Delta m_{CO_2}$ [$\mathrm{Mgha}^{-1}$], in all modeling scenarios. Positive values indicate less $CO_2$ emissions in the blocked scenario. Solid and dashed lines correspond to wet and dry weather conditions, while colors stand for different sets of hydraulic peat properties. Sequestered $CO_2$ was computed using a linear relationship with WTD, Eq.(12).





## 4 Discussion

### 4.1 Comparison to previous studies

A pursuit to estimate the impact of canal blocking in the restoration of tropical peatlands should meet the following four

criteria. First, it should capture the complex and interconnected hydrological processes influencing the block performance. Second, the estimation should be done at spatio-temporal scales similar to the specific scale of the restoration project. Third, a method to meaningfully compare blocked and non-blocked scenarios is needed. Finally, in order to make generalizable claims the sensitivity to weather conditions and peat hydraulic properties should be accounted for.

These criteria are best satisfied with a combination of empirical methods and process-based modelling. If successful, process-

based models are able to combine the relevant physical processes governing water flow at sufficiently large scales. Additionally, they offer a direct method to compare different blocking setups and the ability to study the impact of different weather scenarios and peat hydraulic properties. To our knowledge, the present work, which builds upon our previous study (Urzainki et al., 2020), is the first that tries to meet all the aforementioned criteria.

A few studies have modelled tropical peatland WTD (Cobb et al., 2017; Cobb and Harvey, 2019; Baird et al., 2012, 2017;

Morris et al., 2012; Wösten et al., 2006), but to our knowledge only three have focused on the effectiveness of canal blocks: Jaenicke et al. (2010), Ishii et al. (2016) and Putra et al. (2022). Both Jaenicke et al. (2010) and Ishii et al. (2016) modelled WTD in large areas, and reported that the effect of dams on WTD markedly decayed at 1 km distance from the canals. However, neither of the two works analyzed the effect of different peat hydraulic properties or weather scenarios. The only study that is comparable in scope to ours is Putra et al. (2022). Nevertheless, their study area was several orders of magnitude smaller (20

ha), and their CWL was not dynamic, but manually fixed to field measurements.

The novelty introduced by our discretization of the open channel flow equations is also worth underlining. Unlike traditional methods in which junction nodes require a special treatment (Cunge et al., 1980; Novák, 2010), our approach describes the water flow at all nodes of the computational domain with the same equation, Eq.(A8). Our method is automatically applicable to canal networks of arbitrary topology, thus simplifying the model domain building process.

### 355 4.2 Block impact on WTD

Five main outcomes can be drawn from the present study.

First, dams were, on average and across all the modelled scenarios, beneficial for the rewetting of the study area. This has been observed in many studies in tropical and temperate peatlands (Putra et al., 2021; Sutikno et al., 2019, 2020; Holden et al., 2017; Planas-Clarke et al., 2020; Schimelpfenig et al., 2014; Kasih et al., 2016). Canal blocks are mechanisms that increase

water residence time in the peatland system and, as a result, they can only raise the average WTD. In turn, assuming a strictly positive impact of WTD rise in $CO_2$ emission reduction, this implies that dams must reduce $CO_2$ emissions overall. However, in line with previous studies (Putra et al., 2021, 2022), dams were not enough to maintain WTD above the $-40$ cm limit set by the Indonesian Peatland Regulation Agency everywhere in the peatland during the extremely dry year (see Republic of Indonesia Government Regulation No. 57 Year 2016 about Peatland Ecosystem Protection and Management, 2016).





Second, despite raising the WTD on average, blocks did not do so everywhere in the study area. In some areas downstream from the dams, the WTD was systematically lower in the blocked scenario than in its non-blocked counterpart. To our knowledge, such an effect has not been reported in the literature. Our interpretation of this result is straightforward. The function of dams is to block water, and thus they may reduce water supply to areas with a specific combination of local canal network topology and peat topography.

Third, we found that the impact of dams on WTD was confined to areas close to the blocks. The WTD difference between the blocked and non-blocked scenarios was, on average, 3 cm at 400 m from the canals, and it was 1 mm at a 1 km distance (see Figure 8). Several studies support this claim (Sutikno et al., 2019, 2020; Evans et al., 2019; Putra et al., 2021, 2022; Ishii et al., 2016; Jaenicke et al., 2010). Sutikno et al. (2019), using dipwell measurements, claimed that the radius of action of dams in tropical peatlands is around 170 m, and Ishii et al. (2016) found the modelled WTD rise due to blocks to be about 10 cm at

a distance of 400 m. This result suggests that naively extrapolating WTD measurements performed in the vicinity of the canals may lead to an incorrect assessment of the ability of blocks to raise WTD throughout the study area.

    Fourth, peat hydraulic conductivity had a great impact in the extent to which dams were able to raise the WTD. This effect was also observed in Putra et al. (2022), where blocks had more impact in the higher conductivity site. Peat hydraulic properties govern the dynamics of groundwater flow and, in particular, the conductivity (or, in our model, the transmissivity or the

diffusivity) determines the rate of horizontal water flow (Hillel, 1998; Bear and Cheng, 2010). It follows that higher conductivities result in greater responses of WTD to modifications in the CWL. This implies that peatlands with higher hydraulic conductivities have greater potential for greenhouse gas emission mitigation using canal blocking restoration.

    Fifth, the impact of dams was greater in the dry year than in the wet year. Heavier rainfall events led to smaller differences between the blocked and non-blocked scenarios. Our interpretation is that in the absence of external water input, such as in

prolonged dry periods, the ability of the dams to keep water in the system for a longer time becomes relevant. In contrast, with enough water supply from precipitation, WTD rises regardless of whether dams exist or not. This behaviour differed from the findings of Putra et al. in their small scale empirical and modelling studies (Putra et al., 2021, 2022). Putra et al. reported a decrease in the effectiveness of blocks during dry periods because canals became completely dry and, therefore, blocks were not able to retain any water. In our simulations the majority of the canal network never became completely dry, and thus we did

not observe this effect. However, our simple model of discharge through blocks, Eq.(4), completely restricts water flow through blocks even when the water is below the canal bottom, i.e., when the canal is dry. Thus, even if the canals in our simulation had dried out, our model would not have been able to reproduce the phenomenon observed by Putra et al. The conclusion that can be drawn from the combination of the two studies is the following. The WTD difference between the blocked and non-blocked scenarios is greater in the absence of rainfall. Blocks are likely to delay the point in which canals become dry, yet, once that

point is reached, blocks cease to have any impact on WTD.

### 4.3    Model limitations

Applications of process-based models have two main sources of uncertainty: the simplifying assumptions made in the construction of the model, and the uncertainty in the input data.



The PHM and the CNM use well-established partial differential equations to approximate water flow in peat and in canals.
Despite being two dimensional, the groundwater flow equation of the PHM has been shown to accurately represent water
movement in such domains where the width exceeds the thickness (Connorton, 1985). The two dimensional approximation has
been used extensively in modelling of hydrology in tropical peatlands (Baird et al., 2012; Cobb et al., 2017). When building
the CNM we tested the diffusive wave approximation against the full open channel flow equations (Preissmann method, Cunge
et al. (1980); Haahti et al. (2014)), and concluded that the diffusive wave approximation was accurate enough for this appli-
cation (testing not shown in this study). This was, in part, because the daily timestep was long enough to remove the effect
of the inertial terms present in the full open channel flow equations. As was pointed out previously, the blocks were modelled
as watertight barriers that extend from the surface until the impermeable bottom, while in reality they are relatively porous
(Ochi et al., 2016; Ritzema et al., 2014). The coupling between the PHM and CNM approximates water balance, but it does
not strictly conserve mass. However, this is an unavoidable drawback of any hydrological model that solves groundwater and
surface water flow in different modules (Barthel and Banzhaf, 2016).

The lack of information about the water flow at the area boundaries hindered the choice of appropriate boundary conditions
both in the PHM and the CNM. In particular, there was no data on the water discharge at the catchment outlet, and we set the
boundary conditions in the CWL at that point to no-flow Neumann. This choice, although striking at first, is overrun in the
PHM step by the fixed $-0.2$ m Dirichlet boundary conditions.

Despite being crucial for the understanding of water dynamics, there exist few published measurements of the physical
parameters which govern water flow in tropical peat and in canals. Whereas the variability of the peat hydraulic properties was
taken into account through the different modelling scenarios, the physical parameters governing open channel flow were fixed
in all the scenarios. The parameters specifying canal geometry –width, depth and cross-section shape– were determined by
local expert observations. The block discharge coefficient was determined by adjusting the parameter until the flow rate was
acceptable, and the parameter values for the Manning friction coefficient were adopted from generic values in the literature.
Since the value of these coefficients have a direct influence on water dynamics, more experimental work is needed to correctly
quantify the open channel flow coefficients for tropical peatlands.

All parameters of the model are expected to have some spatial variability. Peat hydraulic properties are known to vary with
vegetation and land use (Baird et al., 2017; Kurnianto et al., 2019); canal geometry changes throughout the canal network
(Ritzema et al., 2014; Ochi et al., 2016); even precipitation and evapotranspiration may change over the study area (Vijith and
Dodge-Wan, 2020). Our model could accommodate all this spatial variability, but in the absence of data we were forced to
assume constant values throughout the study area.

The model validation against the dipwell-measured WTD was limited by the coarse resolution of the PHM computational
domain. The resolution of our digital elevation model was $50$ m $\times$ $50$ m, a scale in which tropical peat surface typically
presents variations of the order of tens of centimetres (Lampela et al., 2016). In the absence of further information about the
precise elevation of the dipwells, we assumed an uncertainty of comparable magnitude in the dipwell WTD measurements.
And since WTD variation at any location was also of tens of centimetres (see Figure 5), this uncertainty prevented any direct
quantitative comparison between the modelled and measured WTD. As a result, we resorted to doing a qualitative estimation





of the model plausibility. Two main features of the reality check of Figure 5 support the validity of the model. First, the model
was unbiased and the range of modelled and measured WTD was comparable. Second, the modelled WTD presented similar
ranges and slopes in the daily dynamics driven by precipitation and evapotranspiration.

Despite the presented limitations, we claim that the modelling setup presented here has a greater potential to study the
rewetting potential of blocks than purely experimental studies have. On the one hand, the effect of some of the aforementioned
uncertainties might be partially compensated by the fact that we have only presented relative comparisons between blocked
and non-blocked scenarios. On the other hand, our model gives theoretically coherent estimates of the dam impact throughout
the area, which is not possible to do with experimental studies alone.

### 4.4  Further study

The present work was limited to the analysis of a single study area, with one block location configuration, for a relatively short
period of time. Future studies might consider how varying the number of dams and their positions affects WTD, since, as we
know, the dam position is critical (Urzainki et al., 2020). Furthermore, the impact of blocks on WTD would probably change in
timescales of decades, which is closer to the typical lifespan of canal blocks (Ritzema et al., 2014; Dohong et al., 2018). When
analyzing long-term scenarios, the effect of climate change in precipitation and evapotranspiration (Gallego-Sala et al., 2018;
Wang et al., 2018; Cai et al., 2014; Pörtner et al., 2022), as well as peat subsidence (Evans et al., 2019; Hoyt et al., 2020; Evans
et al., 2022) will need to be addressed. In order to have a more precise estimate of greenhouse gas emissions, future studies
should take into account emissions of other compounds, such as methane and nitrous oxides. In fact, having shallower WTD
as the only optimization goal may not be desirable due to increased methane emissions (Teh et al., 2017; Wong et al., 2018;
Planas-Clarke et al., 2020; Deshmukh et al., 2020, 2021; Kiuru et al., 2022; Zou et al., 2022; Lestari et al., 2022).

### 5  Conclusions

We modelled the effect that canal block restoration practices had on the WTD of a 22000 ha drained tropical peatland. Our
results show that the blocks raised WTD on average, but their effect was limited. Block impact on WTD at a distance of 1 km
was negligible during one year of simulations, and blocks lowered the WTD in some areas. The effect of dams was largest
during dry periods and in peat soils with higher hydraulic conductivities. We believe that the present modelling setup, which
has been designed with stakeholders' practical management questions in mind, could be adopted by local agencies aiming at a
more effective and evidence-based approach to canal block based peatland restoration.

*Code and data availability.*  The source code and all the data except the DTM, which is property of Deltares, are available at Urzainki (2022).
Forest Carbon provided the data.





*Video supplement.* Animations of $\Delta\bar{\zeta}$ for all modelled scenarios are available as supplementary material.

**Appendix A: Numerical scheme for the diffusive wave approximation**

The open channel flow equations (1) and (2) must be solved in a network of connected channel segments. This connectivity
gives rise to a type of computational node that does not exist when the channel segments are modelled individually: the junction
node, a node shared by more than one individual channel. The traditional discretization of the equations involves writing the
numerical approximations for each individual channel reach first, and then manually adding some mass and energy conservation
conditions at the junction nodes (Cunge et al., 1980; Szymkiewicz, 2010). However, in large and complex channel networks the
traditional approach is tedious and error-prone, because all conservation equations at junctions need to be introduced manually.
In this work we used a slightly different conceptual approach to derive the numerical discretization of the open channel flow
equations that allows to set up the linear system directly from the channel network topology.

The first equation of the open channel flow equations, Eq.(1), is the mass conservation equation. In general, differential
equations that describe conservation laws in one dimension take the form

$$\frac{\partial u}{\partial t} = -\frac{\partial f(u(x,t))}{\partial x}, \tag{A1}$$

where $u$ is the conserved quantity and $f(u)$ is the rate of flow or flux of the conserved quantity.

Let us discretize space and time with regular meshes of width $\Delta x$ and time step $\Delta t$, and define the discrete mesh points
$x_i = i\Delta x$ and $t_n = n\Delta t$, with $i$ and $n \in \mathbb{N}$. Conservative numerical methods are those that can be written as

$$u_i^{n+1} = u_i^n + \frac{\Delta t}{\Delta x}\left[F(u_{i-p-1}^{n+1}, u_{i-p}^{n+1}, \ldots, u_{i+q-1}^{n+1}) - F(u_{i-p}^{n+1}, u_{i-p+1}^{n+1}, \ldots, u_{i+q}^{n+1})\right]. \tag{A2}$$

for implicit schemes, and analogously for explicit schemes (LeVeque, 1992). In the simplest case, $p = 0$ and $q = 1$, this becomes

$$u_i^{n+1} = u_i^n + \frac{\Delta t}{\Delta x}\left[F(u_{i-1}^{n+1}, u_i^{n+1}) - F(u_i^{n+1}, u_{i+1}^{n+1})\right]. \tag{A3}$$

The function $F(u_i, u_{i+1})$, called the numerical flux function, plays the role of the average flux of the conserved quantity $u$,
$f(u)$, between $x_i$ and $x_{i+1}$ during the time interval $[t_n, t_{n+1}]$.

The system of equations arising from the discretization of Eq.(A3) may be interpreted as balance equations at every node.
Indeed, the form of Eq.(A3) ensures that what appears with a plus sign in the equation for $u_i$ must appear with a minus sign in
the equation for $u_{i+1}$. Therefore, the total quantity of the conserved variable $u$ in any region changes only due to flux through
the boundaries.

We may impose this same condition for a general junction node with more than two neighbors. Let us denote the index of
the junction node as $J$, and let $in$ and $out$ be the set of nodes whose flux is incoming and outgoing from $J$, respectively. Then,
the discretized equation for the conservative method at a general junction node is

$$u_J^{n+1} = u_J^n + \frac{\Delta t}{\Delta x}\left[\sum_{k \in in} F\left(u_k^{n+1}, u_J^{n+1}\right) - \sum_{k \in out} F\left(u_J^{n+1}, u_k^{n+1}\right)\right]. \tag{A4}$$



Note that Eq.(A4) reduces to Eq.(A3) for interior nodes. Requiring conservativeness of the numerical scheme at junctions fully specifies the form of the discretized equations. Equation (A4) provides the blueprint for a conservative numerical scheme that is applicable to all nodes in the computational domain.

Our numerical method to solve the mass conservation equation, Eq.(1), is obtained by setting the numerical flux function from node $i$ to node $k$ equal to $Q_{ik}$, the water discharge between those two nodes. The discretization equation for any node in the channel network domain is then

$$h_i^{n+1} = h_i^n + \frac{\Delta t}{B\Delta x}\left[\sum_{k\in in} Q_{ki}^{n+1} - \sum_{k\in out} Q_{ik}^{n+1}\right] + \frac{q_i^{n+1}}{B}. \tag{A5}$$

Note that in our model the channel width $B$ is the same for every node, i.e., $B_i = B$.

The second equation of the diffusive wave approximation, Eq.(2) becomes now useful. It relates the magnitude of the discharge $Q$ to the square root of the gradient of the water elevation,

$$|Q| = C\left|\frac{\partial h}{\partial x}\right|^{1/2}, \tag{A6}$$

where $C = \frac{AR^{2/3}}{n}$.

A straightforward discretization of the spatial derivative results in

$$|Q_{ik}| = C_{ik}\left|\frac{h_i - h_k}{\Delta x}\right|^{1/2}, \tag{A7}$$

where $|Q_{ik}|$ is the magnitude of the water discharge between nodes $i$ and $k$, and $C_{ik} = \frac{1}{2}(C_i + C_k)$.

Finally, we insert Eq.(A7) in Eq.(A5) to get our numerical scheme solving the diffusive wave approximation of the open channel flow equations,

$$h_i^{n+1} = h_i^n - \frac{\Delta t}{B(\Delta x)^{3/2}}\sum_k\left[C_{ik}^{n+1}|h_i^{n+1} - h_k^{n+1}|^{1/2}\mathrm{sign}(h_i^{n+1} - h_k^{n+1})\right] + \frac{q_i^{n+1}}{B}. \tag{A8}$$

The sign function accounts for the direction of the water flow, and the sum in $k$ goes over all neighbouring nodes of the node $i$. As we noted previously, this equation is valid for all nodes in the computational domain, including junction nodes.

With a judicious use of the information about the channel network topology (e.g., by using the adjacency matrix of the graph of canal nodes), this discretization enables a simple implementation of the diffusive wave approximations, since junction nodes do not need to be accounted for separately.

**Appendix B: Parameterization in terms of $\Theta$**

The groundwater flow equation in terms of the hydraulic head $h$ [m] is given by (Urzainki et al., 2020)

$$S_y(h)\frac{\partial h}{\partial t} = \nabla\left(T(h)\nabla h\right) + P - ET, \tag{B1}$$





Where $S_y$ is the specific yield, $T$ is the transmissivity $[\mathrm{m^2 d^{-1}}]$ and $P - ET$ is the difference between the precipitation and evapotranspiration $[\mathrm{m d^{-1}}]$.

Under the change of variables

$$S_y = \frac{\partial \Theta}{\partial h}, \tag{B2}$$

we obtain Eq.(5), the groundwater flow equation in terms of the water storage, $\Theta$ [m]. The water storage is the sum of the volumetric water content $\theta$ $[\mathrm{m^3/m^3}]$ along the soil column,

$$\Theta(h) = \int\limits_{i.b.}^{h} \theta(z) dz, \tag{B3}$$

where $i.b.$ denotes the impermeable bottom.

We can find $\Theta(\zeta)$ by integrating Eq.(B2) and using the parameterization of the specific yield, Eq.(8). We first compute $\Theta(h)$,

$$\Theta(h) = \int S_y dh = \begin{cases} h + C_1 & h > p \\ \frac{s_1}{s_2}\left(e^{s_2(h-p)} - e^{-s_2 p}\right) + C_2 & h \le p \end{cases} \tag{B4}$$

Since $\zeta = h - p$, it is trivial to move between representations in terms of $\zeta$ and $h$.

$$\Theta(\zeta) = \begin{cases} \zeta + p + C_1 & \zeta > 0 \\ \frac{s_1}{s_2}\left(e^{s_2 \zeta} - e^{-s_2 p}\right) + C_2 & \zeta \le 0 \end{cases} \tag{B5}$$

$C_1$ is fixed by imposing the continuity of $\Theta(\zeta)$ at $\zeta = 0$:

$$C_1 = \frac{s_1}{s_2}\left(1 - e^{-s_2 p}\right) + C_2 - p = \Theta_0 - p, \tag{B6}$$

where we have introduced the new parameter $\Theta_0 = \frac{s_1}{s_2}\left(1 - e^{-s_2 p}\right) + C_2$, which is the value of $\Theta$ at the domain threshold $\zeta = 0$.

Different values of $C_2$ only produce a shift in $\Theta$, which is inconsequential for the water dynamics. Here, without any loss of

generality, we choose to set $C_2 = 0$.

Inverting Eq.(B5) we get

$$\zeta(\Theta) = \begin{cases} \Theta - \Theta_0 & \Theta > \Theta_0 \\ \frac{1}{s_2} \ln\left(e^{-s_2 p} + \frac{s_2}{s_1}\Theta\right) & \Theta \le \Theta_0 \end{cases} \tag{B7}$$

The need for $\Theta(\zeta)$ to be invertible poses the hardest constrain on the available possibilities for the parameterization of $S_y(\zeta)$. In particular, with the choice of Eq.(8), note that $S_y$ is independent of the local peat depth, $d$.




Once the map between $\Theta$ and $\zeta$ has been established, we may write the peat hydraulic properties, Eqs.(8) and (9), in terms of $\Theta$.

We find that the specific yield is

$$S_y(\Theta) = \begin{cases} 1 & \Theta > \Theta_0 \\ s_1 e^{-s_2 p} + s_2 \Theta & \Theta \le \Theta_0 \end{cases} \tag{B8}$$

And the transmissivity is given by

$$T(\Theta) = \begin{cases} \alpha(\Theta - \Theta_0) + \beta & \Theta > \Theta_0 \\ t_1 \left( e^{-s_2 p} + \frac{s_2}{s_2}\Theta \right)^{t_2/s_2} - t_1 e^{-t_2 d} & \Theta \le \Theta_0 \end{cases}, \tag{B9}$$


where $d$ is the local depth of the peat profile [m, negative downwards]. Finally, we are in a position to write the diffusivity, which is the only function of the peat hydraulic properties that enters our formulation of the groundwater flow equation,

$$D(\Theta) = \frac{T(\Theta)}{S_y(\Theta)} = \begin{cases} \alpha(\Theta - \Theta_0) + \beta & \Theta > \Theta_0 \\ \dfrac{t_1 \left( e^{-s_2 p} + \frac{s_1}{s_2}\Theta \right)^{t_2/s_2} - t_1 e^{-t_2 d}}{s_1 e^{-s_2 d} + s_2 \Theta} & \Theta \le \Theta_0 \end{cases} \tag{B10}$$

As pointed out in the main text, the diffusivity must be continuous at $\zeta = 0$ (or, equivalently, at $\Theta = \Theta_0$) in order to avoid

numerical instabilities when solving the PDE. Additionally, here we also impose its differentiability at the domain threshold, which may help reduce the instabilities even further. Form these two requirements we find that

$$\begin{cases} \alpha & = \frac{t_1}{s_1^2} \left( t_2 - s_2 + \frac{s_2}{e^{t_2 d}} \right) \\ \beta & = \frac{t_1}{s_1} \left( 1 - e^{-t_2 d} \right) \end{cases} \tag{B11}$$

Note that $D(\Theta)$ is the only function made continuous and differentiable by this choice of $\alpha$ and $\beta$. In particular, $T$ is neither continuous nor differentiable (see Figure 4), and even $D(\zeta)$ is continuous but not differentiable.

*Author contributions.* IU and AL formulated the research goals and methods. IU developed the model code, performed the simulations and prepared the article. MP, HH and AL reviewed and editted the article. SP, JC, OW, PM and RY produced and validated the datasets.

*Competing interests.* The authors declare that they have no conflict of interest

*Acknowledgements.* The authors wish to acknowledge CSC – IT Center for Science, Finland, for computational resources.





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
