# Peer review of "A process-based model for quantifying the effects of canal blocking on water table and CO2 emissions in tropical peatlands"

_Biogeosciences, 2022_

## Referee Comment (RC1)

Tropical peatlands has been drained in the past decades, causing in low water-table conditions, peat decomposition and subsidence, and also greenhouse gases emissions. Drained tropical peatlands are susceptible to fire and smoke, which might lead to health problems.

In order to mitigate environmental issues caused by tropical peatland drainages, massive efforts have been allocated to restore the drained peatlands. Canal blockings are commonly implemented to increase the water table of drained peatlands, but studies evaluating the performance of those structures are limited.

This study simulates the effectiveness of canal blockings in raising the water table of tropical peatlands over large areas, and the estimation of $CO_2$ emission reductions resulted by canal blockings. In this study, a process-based hydrological model was developed to simulate the effect of canal blocks in a 22000 ha of tropical peatlands in Sumatra, Indonesia. The water-table dynamics were modelled in two blocks scenarios (with and without blocks), two El Niño–Southern Oscillation (ENSO) scenarios, and four peat hydraulic properties scenarios. The specific yields and the transmissivity of peat were calculated based on literatures and assumed to be varied with depth, following some empirically generated logarithmic equations.

The simulations were performed using two modules, the canal water flow module (CNM) and the peat water flow module (PHM). The canal water flow module (CNM) was using a diffusive wave approximation of the open channel flow equations. The flow through a canal block was modelled using a coefficient that regulates the flow rate and based on canal water-level state. The peat water flow module (PHM) was using a two dimensional groundwater flow equation. Field water-table data are reported and used to check the model results, though the authors recognized that the variation of peatland topography within the modelling cells (50 m × 50 m) does not allow one-to-one comparison between the modelled and the measured water tables to be done.

The authors have done a novel job. This study accommodated the canal and the peatland water-level interaction in a time step basis, which has not been studied and reported before. I find that this study provides meaningful contribution to tropical peatland studies as it provides efficient approach for analysing tropical peatland water management schemes that involve canal blocks.

Nonetheless, I feel the authors could improve this paper by clarifying some issues as highlighted in this review report.

**General comments:**

1. The time attribute differences between the field data and the model input data must be more clearly summarized and justified in the methodology. For an example, it was mentioned that the simulations were using the data of 1997 (dry year) and 2013 (wet year) (see L214) but the reality check water-table data were using the data of 2020 (see L265).

2. It will be clearer if the author can summarize the canal block performances during dry and wet periods (seasons) in a table, rather than only showing the profile of dry and wet years. Presenting the yearly average of water-table of all modelled scenarios might hide the seasonal performance of canal blocks. The seasonal water-table summary might support the authors' claim that "canal blocks are most effective during dry periods" (see L10).

3. In my opinion, it must be also mentioned and justified that this study has not consider the structured canal distribution density of the study area. The direction and arrangement of the canals were given. Therefore, claims about the limit of canal blocks' effect in refers to the distance from the canal (for example the one in L10) might need to be refocussed. Authors might either ensure that there was only one dominant canal that govern water table behaviour at some transects or admit that several canals might affect the water-table behaviours at some transects.

**Minor issues:**

1. L1-L2: "… among other negative environmental impacts." It needs to be explained or deleted.
2. L2-L3: "These effects can be mitigated by raising the water table depth (WTD) using canal or ditch blocks." Please mention the limit of raising WTD. The statement in L2-L3 is contrast to the statement in L450-L451, which is "In fact, having shallower WTD as the only optimization goal may not be desirable due to increased methane emissions".
3. L7-L8: "The study was performed across two El-Niño Southern Oscillation (ENSO) scenarios, and four different peat hydraulic properties." Please clarify the scenarios. The 1997-Dry year and the 2013-Wet year? Heavily degraded to moderately degraded peatlands?
4. L16: Please put a sentence that summarize the plausibility of the developed model to be implemented by the peatland manager in peatland restoration planning.
5. L21: "…enhancing site productivity…". Please add citations on this if any. Otherwise, please consider to remove it as drainage might not enhance peatland productivity.
6. L30-31: "As a result of the decomposition process, $CO_2$ is emitted, peat subsides, and nutrients are released." It has been mentioned in L23-L27. Consider to combine and simplify the information to avoid redundancy.
7. L44: "Process based models offer a different, complementary approach …" Different than what approach? Please explain.
8. L55-L56: "… a large tropical peatland area …" Please mention the area, 22000 ha?
9. L64-L67: Please add coordinates, for example the lower left corner and the upper right corner coordinates.
10. L74-L76: Please provides the monitoring period of the instruments. Starting on January 22$^{nd}$, 2020?
11. Figure 1: Please use different shapes for Dipwells and Canal blocks. The sentences "Each patrol posts … with monthly frequency." seem redundant (has been mentioned in the text). Delete?
12. Figure 2: Please reformat the symbols (italics).
13. L165: I think Table 2 must be moved here as the specific yield parameterization is discussed here.
14. L176: Do you think that the fixed head Dirichlet boundary of -0.2 m may cause the overall water table conditions never be deeper than -1.5 m (consider that the peat thickness is 5 m)? Please discuss this matter in the discussion section.
15. L212: "… the Sultan Thaha Airport …" Please add coordinates.
16. Figure 3: Legends are incomplete in the first review pdf. Please fix and improve the legends. Mention the used symbols (e.g., c m and m (d)) in the figure's caption.
17. Table 2: The column "Name" can be replaced with "Parameter Scenario", can not it?
18. L230-L233: What is the common height of the blocks? Will it be better if a picture of a block in the field be presented?
19. Figure 4: Please bring it close to Table 2. Some legends of Figure 4 do not complete (some texts are missing). For (b), (c), (d) please use similar range of *x* axis if it is possible, so that the reader can read it clearer and not being confined in a distortion. Please use different line styles for each parameter scenario.
20. L269-L272: Please move or combine this subsection (2.4 Notation) to be under Subsection 2.3 Modelled scenarios.
21. Figure 5: Some legends are not clearly shown. It is better to show some example of modelled scenarios instead of presenting so many lines. It might be better to choose some

output points that are representative to be discussed. Please explain symbols (e.g., m(d), m as d, and m d d) in the figure's caption.

22. Figure 6: The modelling domain needs to be noticeable to the readers. What is the unit of the colour bar? The symbol "$\Delta\bar{\zeta}(m)$ av eraged ov er all scenario s " were not clearly seen/written in this review pdf.

23. Figure 7: Please use different line styles for each scenarios (all 4 of the scenarios). Please make sure that the line styles are noticeable to the readers.

24. L201: Please provides some explanation in Subsection 2.3 about the different perspectives between the distance to canal and the distance to the nearest block. When will each parameter be used to interpret the results?

25. Figure 8: Figures are too small. Please provide some justification in the discussion section about the use of the distance to the nearest block in this spatial study. Please consider that this spatial study involved many canal blocks.

26. Figure 9: Please use different line styles for each scenarios (all 4 of the scenarios). Some axis labels are not completely written.

27. Line 460: The readme file in Urzainki (2022) - Txart block_effectiveness: v1.0.2 was not clear enough in describing the data. It might be better to separate the weather data (P and ET) from the model files/ codes.

---

## Referee Comment (RC2)

**Note regarding water storage gradient for review of doi:10.5194/bg-2022-218**

Alex Cobb, 2022-01-06

In the manuscript, the groundwater flow equation in two dimensions is given as

$$\frac{\partial \Theta}{\partial t} = \nabla(D\nabla\Theta) + P - ET \tag{5}$$

(equation numbers from text). In Appendix B, equation (5) is said to follow from the groundwater flow equation in terms of hydraulic head $h$

$$S_y \frac{\partial h}{\partial t} = \nabla(T\nabla h) + P - ET \tag{B1}$$

based on a change of variables via

$$S_y = \frac{\partial \Theta}{\partial h}. \tag{B2}$$

However, the statements (5) and (B1) are not equivalent because it is not generally true that the horizontal flow $T\nabla h$ can be written $D\nabla\Theta$ with $D = T/S_y$, and with $S_y$ a function of the water table depth, if there are gradients in the surface elevation. An extra term arises from the gradient in surface elevation $p$ combined with the non-uniform profile of specific yield $S_y$, so that $\nabla\Theta \neq S_y\nabla h$.

The specific yield is defined in the text as

$$S_y(\zeta) = \begin{cases} 1 & \zeta > 0 \\ s_1 e^{s_2 \zeta} & \zeta \leq 0, \end{cases} \tag{8}$$

and the water storage $\Theta$ is given in Appendix B by

$$\Theta(\zeta) = \begin{cases} \zeta + \frac{s_1}{s_2}\left(1 - e^{-s_2 p}\right) & \zeta > 0 \\ \frac{s_1}{s_2}\left(e^{s_2\zeta} - e^{-s_2 p}\right) & \zeta \leq 0 \end{cases}$$

(equation B5 in the text, with substitution from equation B6 and the inline equation in the following line), equivalent to

$$\Theta(\zeta) = \int_{\zeta'=-p}^{\zeta} S_y \, d\zeta'.$$

Here is the relevant rule from calculus:

$$\nabla \int_{a(\mathbf{x})}^{b(\mathbf{x})} f(\zeta') \, d\zeta' = f(b)\nabla b - f(a)\nabla a$$

Therefore, the gradient in water storage is computed as

$$\nabla\Theta = \nabla \int_{\zeta'=-p}^{\zeta} S_y(\zeta') \, d\zeta' = S_y(\zeta)\nabla\zeta + S_y(-p)\nabla p$$

Rearranging,

$$\nabla\Theta = S_y(\zeta)\nabla h + [S_y(-p) - S_y(\zeta)]\nabla p.$$

Thus, a term $\left[\frac{S_y(-p)}{S_y(\zeta)} - 1\right]T\nabla p$ is effectively added to the horizontal flow term $T\nabla h$ when equation 5 is used instead of equation B1. This term may be big or small depending on the ratio of the specific yield at the water table and at $z = 0$, and the relative magnitude of the gradients in the water table $h$ and peat surface $p$.

---

## Author Comment (AC1)

In this document we present inline responses to the Reviewers' comments. Their comments are bolded.

The corrections to the mathematical error discovered by Reviewer 2 are discussed in the Appendix.

**1 Reviewer 1**

**1.1 General comments**

**1. The time attribute differences between the field data and the model input data must be more clearly summarized and justified in the methodology. For an example, it was mentioned that the simulations were using the data of 1997 (dry year) and 2013 (wet year) (see L214) but the reality check water-table data were using the data of 2020 (see L265).**

The reality check requires that the input data (weather data) and the reference data (WTD) represent the same conditions, and therefore they must be from the same period. These data were available to us only beginning in January 22nd, 2020, and this was used for the reality check of the study.

Our implicit claim was that, since the model is based in the physics of fluid flow in porous media, if proven to work under a testing environment, it may be extrapolated to other conditions. According to this approach, once the reality check is satisfactory, the model is considered to be plausible in other weather conditions as well.

We had weather records (but no dipwell WTD data) for other time periods, years 1997 an 2013, where we chose to apply the model. (Note as well that the weather data used for the reality check came from several weather stations inside the area, while the data from 1997 and 2013 was gathered from a single weather station outside of the area. The higher-quality of the 2020 data gives additional reasons to choose it for the reality check).

However, we agree with the reviewer that the text could be clearer on this matter. We have added the following two sentences: 1. L246, under the Materials and Methods, was modified to make the year of the used data more explicit: "We performed a reality check of our model by comparing the simulated WTD with the available measured field data (meteorological data from the patrol posts and dipwell data, obtained during the year 2020)." 2. L291, under the Results section for the reality check, the following line was added to make the implicit assumption more explicit: "The model captured the relevant water flow dynamics during the reality check, and thus it is considered to be plausible under the other weather scenarios presented in the study."

**2. It will be clearer if the author can summarize the canal block performances during dry and wet periods (seasons) in a table, rather than only showing the profile of dry and wet years. Presenting the yearly average of water-table of all modelled scenarios might hide the seasonal performance of canal blocks. The seasonal water-table summary might support the authors' claim that "canal blocks are most effective during dry periods" (see L10).**

We think that our claim that "blocks are most effective during dry periods" is well supported by the results we present. By the phrase "dry periods" we mean periods of less rain, regardless of the season of the year. I.e., in our presentation, "dry period" should not be confused with the "dry season", which refers to an annual period of relatively low rainfall within a year (and analogously for "wet periods" versus "wet seasons"). Our results do not depend on the particular definitions of dry and/or wet seasons; rather, they show (see Figure 7) that decreasing the amount of precipitation while leaving other variables constant improves the blocks' rewetting ability.

Therefore, since our intention was not to describe differences between dry and wet seasons, but more generally between dry and wet conditions, we do not think the analysis suggested by the Reviewer is of special relevance. We think that all the information is already contained in Figure 7 and the accompanying text.

Furthermore, it is not clear to us what new insights would be obtained by slicing the results across dry and wet seasons. One of the main of our study is that the comparison between different scenarios begins from the same initial WTD conditions, making the results directly comparable. If we chose an arbitrary point halfway through the year to mark the beginning of the dry season, the starting WTD conditions for that period would not be equal anymore, and drawing conclusions from a direct comparison would not be as straightforward.

Nevertheless, we agree that adding a table summarizing the key results will improve the readability of the text. We will compose a table that captures mean WTD and possibly other relevant metrics out of the results presented in Figure 6 and Figure 7.

**3. In my opinion, it must be also mentioned and justified that this study has not consider the structured canal distribution density of the study area. The direction and arrangement of the canals were given. Therefore, claims about the limit of canal blocks' effect in refers to thedistance from the canal (for example the one in L10) might need to be refocussed. Authors might either ensure that there was only one dominant canal that govern water table behaviour at some transects or admit that several canals might affect the water-table behaviours at some transects.**

We agree with the Reviewer that, due to hydrological connectivity, a general and definitive statement about the rewetting ability of blocks may only be given if we had performed a study with different block setups, and we had controlled the results for a variable amount of blocks. Additionally, block rewetting ability will most likely also depend on time since block construction. Therefore, our study, which only considers one block configuration for a single year of simulated WTD, does not contain enough data to make definitive general claims about block rewetting ability. This is a valid approach, and we tried to implement a similar one in a previous work, [13].

On the other hand, our study area was large and there was a large amount of heterogeneity in the block locations, which capture some of the spectrum of block density present typically in tropical restoration projects: the western part of the catchment, for instance, had more blocks per unit area than the south-eastern part. Indeed, this is exactly what we see in Figure 8: There was a fair amount of variance in the distance up to which blocks affected the WTD.

The scope of the results of our study, then, lies somewhere in the gradient between the two extremes: while it does not have the full explanatory power a fully detailed study that controlled for block

density would provide, it still captures some of the natural variance. We think our discussion gives a fair and balanced description of this fact, and we do not think any changes to the text are necessary.

In other words, and answering the Reviewer's comment more directly, we fully admit that several canals and/or blocks simultaneously affect the WTD at all transects. And we do not think that this fact invalidates the results of the study in any way. This work does not try to isolate the effect that a single block might have in the WTD of a tropical peatland area, but rather to investigate how the WTD might raise in a typical restoration project under different conditions.

It is important to restate here what we say in the manuscript, L372: There are several studies that show a similar WTD difference with respect to distance from the nearest block between blocked and unblocked scenarios [11, 10, 1, 8, 7, 2, 3]. Crucially, this list contains experimental studies which base their claims on direct measurements of WTD before and after building the blocks. We consider that this provides solid reinforcement to our results.

**1.2   Minor issues**

General response about figure rendering issues: There were several comments that addressed poor figure quality. This was an issue with the rendering process on the Biogeosciences site, and was solved after receiving the reviewer's complaints, see the preprint discussion log. Other comments by both Reviewers suggested that some figures should be made larger. We think this may be an artifact caused by the rendering of the files for the discussion phase. We anyway make the commitment to fix the figure sizes during the last typesetting step of the editing process.

**1. L1-L2: "... among other negative environmental impacts." It needs to be explained or deleted.**

We deleted it as suggested.

**2. L2-L3: "These effects can be mitigated by raising the water table depth (WTD) using canal or ditch blocks." Please mention the limit of raising WTD. The statement in L2-L3 is contrast to the statement in L450-L451, which is "In fact, having shallower WTD as the only optimization goal may not be desirable due to increased methane emissions".**

Changed L2-3 to: "These effects can, to a certain extent, be mitigated by ..."

**3. L7-L8: "The study was performed across two El-Niño Southern Oscillation (ENSO) scenarios, and four different peat hydraulic properties." Please clarify the scenarios. The 1997-Dry year and the 2013-Wet year? Heavily degraded to moderately degraded peatlands?**

We agree with the Reviewer that this line does not clearly specify the modelled scenarios. We think that the modelled scenarios do not need to be detailed in the abstract, and we have decided to remove the mention to the peat hydraulic properties.

Furthermore, the mention to the ENSO oscillations might be similarly misleading, and we decided to remove any mention to them here and in Line 219.

The simpler, more clear version of L7-8 now reads: "The study was performed across two contrasting weather scenarios representing dry (1997) and wet (2013) years."

And L219: "The two selected years reflect extremes of the large inter-annual and seasonal variability that are common in the tropics."

**4. L16: Please put a sentence that summarize the plausibility of the developed model to be implemented by the peatland manager in peatland restoration planning.**

Added to the end of the abstract: "We believe that the modelling tools developed in this work could be adopted by local stakeholders aiming at a more effective and evidence-based approach to canal block based peatland restoration."

**5. L21: "...enhancing site productivity...". Please add citations on this if any. Otherwise, please consider to remove it as drainage might not enhance peatland productivity.**

Here we are adopting the definition of site productivity given by [9]: "Site productivity is a quantitative estimate of the potential of a site to produce plant biomass (...). The term site productivity is often used in a more narrow sense to refer to that part of the site potential that is or is expected to be realized by the trees for wood production."

It is well established that when drainage removes excess water from the surface of the peat, it improves physical and nutrient supply for vegetation growing conditions thus improving site productivity. For a recent review, see [6] and the references therein. Also worth mentioning are the works by some of the authors, [4, 5].

Thus, we don't think our text needs any improvements in this point.

**6. L30-31: "As a result of the decomposition process, CO2 is emitted, peat subsides, and nutrients are released." It has been mentioned in L23-L27. Consider to combine and simplify the information to avoid redundancy.**

We do not think that there is any redundancy. The statements in L23- L27 are factual, while L30-L31 describe the processes which lead to those outcomes. Nevertheless, we propose changing L30-L31 to "It is as a result of the decomposition process that CO2 is emitted, peat subsides, and nutrients are released.", to underline that logical relationship.

**7. L44: "Process based models offer a different, complementary approach..." Different than what approach? Please explain.**

L44 refers to the drawbacks of relying solely on empirical measurements that are localized close to canals, which is the focus of the previous paragraph. We do not think this needs any further clarification.

**8. L55-L56: "... a large tropical peatland area ..." Please mention the area, 22000**

**ha?**

Changed L55 to "... a large tropical peatland area (22000 ha)"

**9. L64-L67: Please add coordinates, for example the lower left corner and the upper right corner coordinates.**

We will add these coordinates in the final version.

**10. L74-L76: Please provides the monitoring period of the instruments. Starting on January 22nd, 2020?**

Added the following line after L76: "Weather and WTD data was collected for 365 days, starting from January 22nd, 2020."

**11. Figure 1: Please use different shapes for Dipwells and Canal blocks. The sentences "Each patrol posts . . . with monthly frequency." seem redundant (has been mentioned in the text). Delete?**

A "patrol post" is not a common or well-defined concept. As such, we prefer to repeat here what the Figure legend stands for.

Reviewer 2 asked to restate the DTM resolution in the Figure caption, a somewhat contradictory requirement to the suggestion by this reviewer. When in doubt, we prefer to be redundant in the information displayed in the Figures and the text, but we will change our stance if the Editor considers that eliminating redundancies is the right course of action.

**12. Figure 2: Please reformat the symbols (italics).**

We will try our best to italicize the symbols.

**13. L165: I think Table 2 must be moved here as the specific yield parameterization is discussed here.**

This is escapes our possibilities. The typesetting strategy in LaTeX is to "float" the figures and tables, i.e., to let the software decide on their best placements.

**14. L176: Do you think that the fixed head Dirichlet boundary of -0.2 m may cause the overall water table conditions never be deeper than -1.5 m (consider that the peat thickness is 5 m)? Please discuss this matter in the discussion section.**

A similar concern was raised by Reviewer 2. He asked: "[Would it be] Possible to state somewhere the sensitivity of model outputs to this choice of water table at the boundaries?" This is an answer to both comments.

It is the case with any boundary conditions that the accuracy of the simulations decreases towards the domain boundary. It seems plausible to us that the distance to which the signal of the Dirichlet boundary condition can propagate is similar to the distance to which a change in the CWL affects

WTD. Given our results in the present study (cf. Figure 8), we can estimate that inaccuracies introduced by the Dirichlet boundary conditions would only affect the WTD in a buffer zone of up to 1 km.

Anyway, we think these inaccuracies do not play a big role in the results and conclusions of our study. Our study was framed so that all the results presented (Figures 6-9) are relative, i.e., they are based on WTD differences between blocked and unblocked scenarios. Indeed, taking a look at the WTD difference maps shown in Figure 6 and 8, we see that the difference between the blocked and unblocked scenarios is negligible at the domain boundaries. Therefore, we can conclude that most of the inaccuracies introduced by our choice of boundary conditions are "subtracted away" when the comparison between scenarios is made.

We agree with the two Reviewers that this idea should be more clearly stated. We added the following paragraph to the Model Limitations section of the Discussion, at L414:

"The Dirichlet boundary conditions are themselves a source of inaccuracy in the simulated WTD, as would be the case with any choice of boundary conditions. It seems plausible to us that the distance to which the signal of the Dirichlet boundary conditions can propagate may be similar to the distance to which a change in the CWL affects WTD. Given our results in the present study, we can estimate that inaccuracies introduced by the Dirichlet boundary conditions would only affect the WTD up to a 1 km buffer zone around the boundaries of the study area. We nevertheless argue that these inaccuracies do not undermine the results presented in this work because they are based on direct comparisons between different scenarios. Indeed, the WTD difference maps displayed in Figures 6 and 8 (b) and (c) confirms that the difference between scenarios is negligible at the study are boundaries. It is reasonable to think then that most of the error introduced by the boundary conditions does not have a large effect in our results."

**15. L212: "... the Sultan Thaha Airport ..." Please add coordinates.**

Added (1°38'01"S 103°38'24"E) to L212.

**16. Figure 3: Legends are incomplete in the first review pdf. Please fix and improve the legends. Mention the used symbols (e.g., c m and m (d)) in the figure's caption.**

See comments about figure rendering above.

**17. Table 2: The column "Name" can be replaced with "Parameter Scenario", can not it?**

Column name changed to "Parameter scenario".

**18. L230-L233: What is the common height of the blocks? Will it be better if a picture of a block in the field be presented?**

Added after L233: "The typical block is made out of surrounding peat, and covers the canal width entirely up to the local peat surface."

**19. Figure 4: Please bring it close to Table 2. Some legends of Figure 4 do not**

**complete (some texts are missing). For (b), (c), (d) please use similar range of x axis if it is possible, so that the reader can read it clearer and not being confined in a distortion. Please use different line styles for each parameter scenario.**

As explained above, we cannot decide on the location of figures and tables. For the Figure 4 rendering problems, see above. It is difficult to use the same x axis in (b) and (c) because $K = t_2 T$, so that it is an order of magnitude greater. It also makes little sense, in our opinion, given the size difference with panel (d), that this also has to have the same axis range.

We also disagree about the linestyles. We carefully chose to maintain the same linestyles (line colors for different parameter scenarios, dashed lines for dry conditions, solid lines for wet conditions) throughout the manuscript (Figures 7, 8 and 9) to improve the reader's experience. Now, changing the linestyles in Figure 4 would mean also changing the styles in all the rest of the figures. As we explain below, the readability of the figures, pariculary Figure 7, would suffer from such change.

We appreciate the advice, but we prefer to keep it as is.

**20. L269-L272: Please move or combine this subsection (2.4 Notation) to be under Subsection 2.3 Modelled scenarios.**

We know that the brevity and the straightforwardness of section 2.4 is a bit awkward, but we didn't find any other more logical placement. We do not agree with the Reviewer that remarks about notation fall under a description about modelled scenarios. Therefore, with all due respect, we are leaving it as is.

**21. Figure 5: Some legends are not clearly shown. It is better to show some example of modelled scenarios instead of presenting so many lines. It might be better to choose some 3output points that are representative to be discussed. Please explain symbols (e.g., m(d), m as d, and m d d) in the figure's caption.**

Comments about figure rendering above apply here. The difficulty to interpret Figure 5 was a common complaint by the two Reviewers. We will attempt to improve the figure by separating the modelled and measured datapoints in some way.

**22. Figure 6: The modelling domain needs to be noticeable to the readers. What is the unit of the colour bar? The symbol "Δ"ζ(m) av eraged ov er all scenario s " were not clearly seen/written in this review pdf.**

Comments about figure rendering above apply here. We do not know whether the Reviewer means that the finite element method mesh cell should be visible. This was related to a comment by Reviewer 2. For the next version, we will try to add a subfigure showing the mesh, also containing a diagram explaining its relationship with the canal network.

**23. Figure 7: Please use different line styles for each scenarios (all 4 of the scenarios). Please make sure that the line styles are noticeable to the readers.**

Figure 7 has 8 lines (2x 4 params). We tried to add one line style per category, but we found that it clutters the Figure, which already contains features which are hard to see. In fact, Reviewer 2 had

a comment about the visibility of Figure 7, see below.

**24. L201: Please provides some explanation in Subsection 2.3 about the different perspectives between the distance to canal and the distance to the nearest block. When will each parameter be used to interpret the results?**

We are not sure what the reviewer exactly means with this question. Blocks affect CWL, and therefore the two quantities mentioned by the reviewer are related and inseparable. When analyzing the spatial properties of the simulated scenarios we always used the distance to nearest block, as can be seen in Figure 8. Comparing Figures 1 and 6 might clarify why we preferred this quantity over the distance to canal. As can be seen in Figure 1, there were some canals in the study area which were not blocked at all, e.g., the canals in the south-east part. The WTD difference between the blocked and unblocked scenarios at those canals, as shown in Figure 6, was negligible, as expected. Thus, using distance to canals instead of distance to blocks as the quantity to analyze would have introduced some noise in the results.

We do not think that the text should be changed over this point.

**25. Figure 8: Figures are too small. Please provide some justification in the discussion section about the use of the distance to the nearest block in this spatial study. Please consider that this spatial study involved many canal blocks.**

As far as our understanding of the Reviewer's comment goes, this point is closely related to the third General Comment discussed above. To summarize our previous point: it is outside of the scope of this study to try to separate the effect of a single block in the WTD. Rather, we investigate the WTD difference that canal blocks may make in a typical restoration project under different conditions.

We do not think that the Discussion about this issue needs any amendments.

**26. Figure 9: Please use different line styles for each scenarios (all 4 of the scenarios). Some axis labels are not completely written.**

See above for the discussion of the linestyles. See also our comments about figure rendering.

**27. Line 460: The readme file in Urzainki (2022) - Txart block-effectiveness: v1.0.2 was not clear enough in describing the data. It might be better to separate the weather data (P and ET) from the model files/ codes.**

The Reviewer is right: we forgot to include the precipitation and evapotranspiration data. We thank the Reviewer for the notice. We will upload it.

**2 Reviewer 2**

**2.1 Specific comments**

**- Title: I suggest dropping the word "restored," at the authors' discretion. Though canal blocking is a restoration activity, the results presented here (and elsewhere) suggest that it would be premature to say that a peatland in which canals have been blocked has been "restored." Anyway, the restoration activity is implied by the phrase "canal blocking" in the title.**

The word "restored" was dropped from the title.

**- Including the phrase "peatland restoration" in the abstract might make it easier for people to find the article by keyword searches.**

Thank you for the advice. We have changed one occurrence of "raising water table" for the more general "peatland restoration" in the abstract. We will also add it as a keyword in the manuscript publication process.

**- Lines 11-12: "blocks raised the annual mean WTD by only 0.9 cm." This spatially-averaged amount is small, but perhaps the water table is raised the most where it is most important, near canals? This wouldn't matter much for CO2 fluxes if they are an affine function of water table depth, but it could affect the likelihood of fire. This is something that could be mentioned in the Discussion, at the authors' discretion.**

We will add a line to this effect under "4.2 Block impact on WTD" (around L370).

**- Line 20: "extensive tropical peatland areas have been converted to agricultural and plantation forest production" Might consider also citing this paper, which focuses precisely on this issue: Miettinen, J., Shi, C., and Liew, S. C. (2016). Land cover distribution in the peatlands of Peninsular Malaysia, Sumatra and Borneo in 2015 with changes since 1990. Global Ecology and Conservation, 6, 67–78. doi:10.1016/j.gecco.2016.02.004**

We intended to cite this work, but for some reason –likely because we confused it with the cited Miettinen 2017 reference– it did not make the final version. We have included it now. Thank you very much for bringing this to our attention.

**- Line 26: "export to water courses": change to "export of nutrients to water courses" toavoid any ambiguity**

done.

**- Lines 24-26: I think all the references in this paragraph except Laurén et al., 2021a and Nieminen et al., 2017 report findings from tropical peatlands. The inclusion of these references deviates from the paragraph's focus on tropical peatlands; are they needed?**

Citations removed.

**- Line 50-51: "Putra et al. (2022), on the other hand, presented a good experimental setup to analyze block efficiency" This study also examined the effects of bunds; this could be mentioned here or in the Discussion.**

Changed to "Putra et al. (2022), on the other hand, presented a good experimental setup to analyze block and bund efficiency"

**- Figure 1: The symbols in the legend and the main figure are hard to see. The text is also a bit small at printed size. Could the whole figure be enlarged, or at least the symbols and text enlarged? It would be helpful to restate the resolution of the DTM here. It would also be nice to see some more detail on the DTM in the vicinity of the canals (a zoom-in or cross-section?). Are depressions around canals visible in the DTM?**

The figure will be enlarged, if necessary, in the last edition step. We added mention in the figure caption about the DTM resolution (50 m x 50 m). We agree that such information would be useful. We will add an additional panel to Figure 1.

**- Lines 79-80: "At each timestep, these modules work in an alternate fashion to update the next day's canal water level (CWL, meters, negative downwards) and WTD across the study area." Not a requirement for publication, just a comment: for future work, it would be very interesting to see error estimation (and possibly control) linked to this alternating-step approach. The following paper that the authors may have seen presents a clever and very general approach, based on Richardson extrapolation, to error estimation and correction of operator-splitting methods: Gasda, S. E., Farthing, M. W., Kees, C. E., and Miller, C. T. (2011). Adaptive split-operator methods for modeling transport phenomena in porous medium systems. Advances in Water Resources, 34(10), 1268–1282. doi:10.1016/j.advwatres.2011.06.004**

We were not aware of this work. We agree with the reviewer in the importance both of estimating the operator-splitting error, and the interest of the cited paper. We will add a section that mentions this venue of research in the Discussion.

**- Line 114: "The Manning friction coefficient was described as follows"Briefly state the basis or precedent for the functional form of n when the water surface is above the canal bed.**

We consider that Lines 109–113 do exactly what the reviewer is asking for. We mention that i) $n$ decreases as it reaches the channel bed, ii) it does so non-linearly, and iii) $n$ above the channel bed must be orders of magnitude higher than below, to account for peat porosity. That is why we only made a small clarification in Line 114: "Therefore, the Manning friction ...".

**- Lines 118-119: "In the absence of better information sources the values for these parameters were chosen by trial and error" Clarify - how were the outcomes evaluated when choosing parameters?**

We experimented with parameter values that resulted in parameter values within the ranges presented in [12]. They all resulted in plausible values for "natural looking" CWL with a full channel, so we chose one value in the middle of the range. The available data (daily point measurements of CWL in sparse areas in the channel network) were not enough to discriminate between the different parameter values.

Lines 118-120 were simplified to the following text: "In the absence of better information sources the values for these parameters were chosen so that the value of $n$ when the canal is full of water was $n = 0.055m^{-1/3}s$, which is in the range described in Szymkiewicz (2010), and the value for flows below the canal bed was $n = 100m^{-1/3}s$ (see Table 1)."

**- Lines 135-138: "In our implementation of the model, disconnected components of the canal network were solved independently, in parallel processes. The accelerated Newton- Raphson method introduced by Liu and Hodges (2014) was used to solve the resulting linear system of equations." If I understand correctly, should this not read "systems of equations" instead of "system" (one system for each connected component), and "nonlinear" instead of "linear" (each solver step solves a linear system, but each system of equations being solved is nonlinear)? If I misunderstood - please clarify.**

The understanding is correct. There is one system of nonlinear equations per separated channel network graph component. Of course, the Newton method works by linearizing the nonlinear equations –so one may argue the Newton method solves linear equations. However, we agree with the reviewer that changing "linear" to "nonlinear" might help in improving the understanding of the underlying mathematical setup. However, one Newton method is applied per disconnected graph component (as opposed to a single system of equations for the whole channel network), and since parallel processes are mentioned in the previous sentence, we think that leaving "system of equations" makes the point clearer.

**- Line 144, equation 5: It is not generally true that D grad Θ = T grad h with Sy and Θ as defined here; there is an additional term arising from the gradient in the peat surface elevation and nonuniform specific yield. Please see the PDF attached to this review.**

We agree that there is a mistake. This issue is dealt with in the Appendix.

**- Line 166, Equation 9: T, $\alpha$, and $\beta$ are all functions of d, which varies in space; is it worth highlighting this by including this functional dependence in the equation by writing $T(\zeta, d)$, $\alpha(d)$ and $\beta(d)$ here and / or in equation 10?**

Inserted $T(\zeta, d)$ in Eq.(9); added dependence $\alpha(d)$ and $\beta(d)$ in Eq.(10). Note that these expressions may change with the fixed parameterization arising from the mathematical error spotted by the reviewer. In any case, dependencies on the spatially changing variables $p$ and $d$ will be made explicit in future versions.

**- Line 167: "where d is the local peat depth [m, negative downwards]" Wording (with "negative downwards") a little confusing to me; consider "where d is the local peat thickness [m]"?**

We think that both wordings have some merit to them. "depth [m, negative downwards]" underlines the sign, whereas "peat thickness [m]" is more natural, but does not carry information about what sign should $d$ have. However, we accept the Reviewer's criterion and make the proposed changes, as the sign of $d$ might be inferred from the equations.

**- Lines 175–175: "Fixed head Dirichlet boundary conditions with value $\zeta = -0.2m$ were applied at the domain boundaries."Possible to state somewhere the sensitivity of model outputs to this choice of water table at the boundaries?**

See response given to the Reviewer 1 about the same issue.

**- Lines 178-186, Module interaction "water flow between any two adjacent mesh cells that corresponded to the canal network was completely restricted by setting D = 0 in the cell faces." Does this refer to cell faces completely within the canal? not cell faces along the boundary of the canal? "the head difference at canal cells before and after the execution of the PHM was used to compute the lateral water inflow q for each timestep" Does this refer to the gradient towards the canal, evaluated as a finite difference across faces along the boundary of the canal? A very simple drawing showing the mesh and flows in the neighborhood of a short section of canal could help a lot to understand exactly what was done.**

The first sentence refers to cell faces completely within the canal.

The second sentence refers to the difference in WTD at canal mesh cells between two different timesteps. This is to say: $q(t+1) = \frac{1}{\Delta t}[WTD(t+1) - WTD(t)]$, where $WTD$ referes to the water table (or water level) at the canals.

We agree with the Reviewer that such a Figure would improve the readability of the text. We will add it to Figure 1.

**- Lines 188-189: "Whenever the CWL rose above ground, the volume of ponding water was distributed instantaneously throughout the cell area in the PHM step" More explanation about how this was done would be helpful. What is "the cell area" in this statement? Was water allowed to spread to cells some distance away from the canal, depending on the peat surface elevations in the vicinity of the canal?**

It must be kept in mind that there are two different domains for the CNM and the PHM. The CNM is a 1D traditional finite differences. The PHM is a 2D triangular mesh. The CNM domain is embedded within the PHM mesh, so that there are mesh cells that correspond to the canal network. When the canal water level goes above the surface in the CNM domain by $x$ metres, the water should not raise everywhere in the corresponding mesh cell by $x$ metres, but rather, it will naturally be spread out. (If we allowed such behaviour, the water mass balance would be violated, because new water would be "created" around the canals) The "cell-area", then, refers to the area of the canal mesh cell. Thus, answering the last question, no, the water did not spread to some cells around the canal –just to the same mesh cell.

We realize this is a subtle point. We think that the additional panel that we will add to figure 1 (see comment above) will help in understanding this point more clearly.

**- Line 195: Minor point, for accuracy, was the lidar dataset collected by Deltares as stated? I know that in many cases they have analyzed data (and created DTMs) based on LiDAR datasets collected by third parties.**

The impression we got from our interactions with Deltares is that the LiDAR dataset was indeed collected and analyzed by them.

**- Line 207, "reality check": What was done in the reality check? Were results examined visually?- Figure 3: Units on left vertical axis: use "mm / d" for clarify, as in Figure 7?**

L207: Added "... in which we visually compared the model results". We will update the units of Figure 3 to mm/d.

**- Line 239: "In an Indonesian peatland, Cobb et al. (2017)..." The site is in Brunei, so this could be "In a Brunei peatland" or "In a Borneo peatland"**

Corrected.

**- Lines 248-249: "This small scale variation in peat elevation is known to be of tens of centimeters (Lampela et al., 2016)" For what it's worth, this is also shown in the total station transect survey presented in Cobb et al., 2017 (Figure 3); that paper could also be cited here, if desired (given that it is cited anyway).**

Reference added.

**- Figure 4: Specific yield function: Not necessary for publication, no need to cite this paper or discuss this issue, but possibly of interest: Dettmann and Bechtold (2016) and references therein discuss the effects of microrelief on the specific yield profile near the ground surface. This study also discusses the effects of the truncation of the soil moisture profile at the soil surface on the specific yield, which, in a homogeneous soil, results in a decrease in the specific yield as the water table approaches the surface from below.**

Interesting and unknown study for us; we thank the reviewer for sharing.

**- Line 266: "The modelled WTD raster that best agreed with those sensor measurements was selected as the initial condition..." What is the set of simulations that are being chosen from here? Do I understand correctly that the initial-conditions raster was chosen from among the 50 rasters from the end of each of the 50 simulation days? Please clarify. How was agreement evaluated?**

Yes, the Reviewer's understanding is correct. We changed the previous sentence to make this point more clear. L265 now reads "We then compared the WTD at the dipwell locations for each of the 50 rasters with the dipwell measurements from January 22nd, 2020, the first sensor measurements of the year."

Mean squared error was used to evaluate the agreement. L266 changed to "The modelled WTD

raster that resulted in the smallest mean squared error was selected as the initial condition for all scenarios."

**- Line 288-289: "The daily fluctuations produced by variations in precipitation and evapotranspiration were comparable in their magnitude."I wasn't sure what was meant here.**

We meant that the magnitude of the WTD rise after heavy rainfalls and the WTD drop in the recession periods was similar in magnitude in the modelled and measured. This, in turn, suggests that the specific yield is correct.

We agree that our wording was not precise enough. We changed this sentence to: "The magnitude of the WTD rise after heavy rainfalls and the WTD drop in the recession periods were similar in magnitude in the modelled and measured datasets."

**- Figure 5: The measured WTD is lost behind the modeled lines. It might help if the measured WTD were plotted on top, and made opaque or darker? It might be easier to follow the modeled WTD if it is plotted in back, because it is represented by lines.**

We are not sure about the origin of those points. It can be a wrong datum, or it can be local depressions, unseeable from the DTM, that are canalizing the water to that spot.

**- Figure 5: In my PDF viewer (evince) and on my printer, parts of the axis labels in this and some of the other plots are missing or replaced by triangles, so I couldn't read the axis labels in this figure, though I could guess at what they should be from context.**

This is likely related to the figure rendering issues described before, see above.

**- Figure 5: It looks like high WTDs at some locations in the observed dataset are not being captured in the model. Or is this somehow an artifact of the DTM resolution, or the datum for some of the manual measurements?**

We are not sure about the origin of those points. It can be a wrong datum, or it can be local depressions, unnoticeable from the DTM, that are canalizing the water to that spot. See the related comment below for the changes we introduced in the text.

**- Figure 6: It would be nice to see the canal block positions in this figure too if this does not clutter it too much; it could help a reader to immediately interpret the areas where the water table was lower in the blocked scenario.**

We will try to add them. But, as the Reviewer mentions, they might clutter the figure. If so, we will not include them.

**- Lines 323-324: "... higher hydraulic conductivities had the effect of increasing the spatial extent of the influence of blocks." The higher conductivities correspond to different K and T profiles, so I wonder how the mean water tables were different for the different parameterizations without canal blocks? I guess that the water table will reside lower in the peat profile in simulations using the higher K parameterizations?**

The reviewer's guess is right. This information is already contained in Figure 7, although we could be more explicit about it. Following also Reviewer 1's advice, we will add a Table summarizing the results shown in Figures 6 and 7.

**- The motivation for the different parameterizations was not very clear to me; perhaps more could be said about this in the Introduction or the Discussion. Is this an exploration of the sensitivity of the results to a these physical properties, because they are not very well constrained by calibration or validation?**

The Reviewer is right in that the peat hydraulic properties are not well constrained by calibration or validation. It is well known in the literature, as we point out in the study, that the peat hydraulic properties differ a lot between sites, and may as well differ within the same study area. This being the case, even if we would have managed to measure the exact specific yield and transmissivity of the peat in our study area, it would still be interesting to run our modelling study with different properties in order to see their effect. It is in this sense that studying several different hydraulic peat properties helps to generalize our results.

This is very much related to a sensitivity analysis, but it is not exactly that. Since the term "sensitivity analysis" has a specific, rigorous definition, we chose not to use it in our manuscript.

**- Figure 7: The dashed lines and shaded areas were only visible to me at 200% zoom. Could the figure be made larger (full page width)?**

We admit that the shaded areas are rather small, but we think it is more important to keep all subfigures within the same vertical axis range. In fact, it was because the shaded region was small that we decided to plot the bottom subfigures, Figures 7 (c) and (d).

Figure 7 is already full-page width. We hope the figure will be rendered a bit larger once the publication template is used. If not, we will try to improve this and any other figures during the typesetting phase of article editing process.

**- Figure 7 caption: "Spatially averaged WTD differences between the blocked and non-blocked scenarios for all the modelled scenarios": slightly confusing because the first rows are WTDs, not WTD differences. Perhaps "Spatially averaged WTD and WTD differences..."**

Changed caption to: "Spatially averaged WTD (top) and WTD differences (bottom) ..."

**- Figure 8: Good figure but too small to see without zooming way in (not sure if this has to do with how the PDF is created for review).**

The image is already full-page width. We suspect it has to do with the review-template version. We will take care of all figure visibility details, if needed, in the typesetting step of the publication process.

**- Lines 337-338: "Finally, in order to make generalizable claims the sensitivity to weather conditions and peat hydraulic properties should be accounted for." How do I use the peat hydraulic properties to generalize?**

See the response above about the sense in which analyzing different parameters leads to some generalization.

**- Lines 342-343: "To our knowledge, the present work, which builds upon our previous study (Urzainki et al., 2020), is the first that tries to meet all the aforementioned criteria." Is this true of studies of higher-latitude peatlands as well? For example, there have been a lot of outputs from rewetting studies in UK peatlands in recent years; I am not sure if any of them meet all the criteria stated here. Clarify that these statements are meant to apply to tropical peatlands (rather than peatlands generally), if that is the case? Also, though the approach is different, this recent study might also be worth mentioning in this section: Salehi Hikouei, I., Eshleman, K. N., Saharjo, B. H., Graham, L. L. B., Applegate, G., and Cochrane, M. A. (2023). Using machine learning algorithms to predict groundwater levels in Indonesian tropical peatlands. Science of The Total Environment, 857, 159701. doi:10.1016/j.scitotenv.2022.159701**

L343: We added the qualifier "... is the first that tries to meet all the aforementioned criteria in tropical peatlands".

We disagree with the reviewer in this point. In our view, the remote sensing-based water table depth models solve some of the problems of traditional block efficiency assessment methods. Namely, it solves the issue of capturing WTD changes spatiotemporally, over large enough scales (as opposed to dipwell point-measurements in small spatial regions). However, crucially, they lack the ability to directly compare WTD in different block configurations. Our argument is that given the meteorological variability in the tropics, comparisons of WTD in a given area before and after building blocks might not give any information about the actual performance of the blocks, since comparable WTD variations are possible just with natural rainfall fluctuations. In other words, any measurement setup, including remote-sensing techniques, would need large temporal scales and a large sample size to account for rainfall variability. While they might have other shortcomings, process-based models are able to directly compare hypothetical situations by varying one parameter of the model at a time – in this case, the presence or absence of blocks – and to estimate their impact on WTD.

However, we agree that mentioning the impressive recent advancements in remote sensing techniques for WTD measurement is a good idea, and we will add a reference to those under "4.4 Further study" section.

**- Lines 407-410: "The coupling between the PHM and CNM approximates water balance, but it does not strictly conserve mass. However, this is an unavoidable drawback of any hydrological model that solves groundwater and surface water flow in different modules (Barthel and Banzhaf, 2016)." As mentioned above - the approach in Gasda et al. (2011) could make it possible, in future work, to quantify and control the error from coupling.**

We will add the Gasda et al.(2011) reference here.

**- Lines 415-423: "... The parameters specifying canal geometry –width, depth and cross- section shape– were determined by local expert observation..." Some of the explanation here of how parameters were set should also appear in the Methods.**

We will move some of those to the Materials and Methods section.

**- Line 435: "the modelled WTD presented similar ranges and slopes in the daily dynamics driven by precipitation and evapotranspiration." This could be a place to mention the disagreement in measured and modeled water table depth distributions at higher water tables (see comment on Figure 5).**

We preferred to add the following line right before that, in Line 434:

"It should also be mentioned that we are not sure about the origin of the dipwell measurements showing WTD up to 1 m above the surface in Figure 5. It could be due to a wrong datum, or to local depressions that are unnoticeable in the DTM which are canalizing the water to certain spots."

**- Line 451-452: "In fact, having shallower WTD as the only optimization goal may not be desirable due to increased methane emissions" Though I'm not aware of any tropical peatland study where methane emissions outweigh CO2? Worth mentioning this?**

L451 changed to: "In fact, having shallower WTD as the only optimization goal may not be desirable due to increased methane emissions—although we are not aware of any studies where methane emissions have been shown to surpass $CO_2$ emissions"

**- Line 460: "Forest Carbon provided the data." Perhaps "Forest Carbon Pte. Ltd." or a URL to make it easier to find the organization, for those who have not read the affiliations at the beginning of the article?**

L460: Added Pte. Ltd. and a url as suggested.

**- Line 517, Equation B1: Sy is a function of $\zeta$ or of h and p (equation 8); T is a function of d as well as h (equation 9). Write $Sy(\zeta)$ or Sy(h, p) and $T(\zeta, d)$ or T(h, p, d)?**

All the dependences with $d$ an $p$ have been made explicit in the corrected version of the parameterizations, and they will be written in the Appendix in the next version of the manuscript. See separate .pdf file for these improvements.

**- Lines 524-525: Equation B3, "where i.b. denotes the impermeable bottom": I believe the lower bound of the integration needs to be at the fixed vertical datum z = 0 for consistency with the definition of $\Theta$ in equations B4 and B5.**

This is fixed in the separate .pdf document.

**- Line 527, equation B4: Show the limits of integration, which I believe should be z = 0 to h.**

Same as above. This is fixed in the separate .pdf document.

**- Lines 533 and 537, equations B6 and B7: Is it worth writing $\Theta_0(p)$ to highlight the functional dependence of $\Theta$ on p, which varies in space? Technical corrections**

Same as above. This is fixed in the separate .pdf document.

**2.2 Technical corrections**

**- Line 28: "onsets mechanisms": Initiates? Activates?**

L28 changed to "activates"

**- Line 120: "rage" -¿ "range"**

Corrected.

**- Line 199: Missing a verb at "scenarios, the following section for more details"**

L199 changed to "... scenarios, see the following section for more details."

**- Line 236: "The values reported for the transmissivity and its derivative, the conductivity, K, vary ostensibly" - "ostensibly" does not seem the right word, given the context? If this is really what was meant, explain more?**

L236 Changed to "vary significantly".

**- Line 538: "constrain" -¿ "constraint"**

Corrected.

**A   Appendix: Mathematical error**

**A.1   The error**

The second reviewer pointed out that there is an error in the derivation of the groundwater flow equation in terms of $\Theta$. I agree with his criticism. In this document I explain what our original manuscript overlooked and I provide the correct expressions. The approach I follow here is slightly different from the one followed by the reviewer. As I show in the last section, both are equivalent. I chose the approach that follows because it made both the problem and the solution more clear.

The $h$ based form of the groundwater flow equation, as presented in the text, is

$$S_y(h)\frac{\partial h}{\partial t} = \nabla[T(h)\nabla h] + P - ET \tag{1}$$

This expression is already setting a foot in the wrong direction. The reason is that our parameterization of $S_y$ did not only depend on $h(\vec{x}, t)$, but on $\zeta(\vec{x}, t) = h(\vec{x}, t) - p(\vec{x})$, as the reviewer pointed out.

Therefore, $\Theta$ must also depend on $h$ and $p$ at least. This previously hidden dependencies on other spatially varying functions adds an extra term to the groundwater flow equation, as the reviewer explained.

After our review, we consider that in order to interpret $\Theta$ as a measure of the amount of water of the peat column, we should include also a dependence on the depth of the peat column, $d(\vec{x})$. This adds one further term to the groundwater flow equation.

**A.2 Corrected expressions**

**A.2.1 Derivation of the groundwater flow equation in terms of $\Theta$**

The transformation we are seeking is of the form

$$S_y(\zeta)\frac{\partial h}{\partial t} = \frac{\partial \Theta}{\partial t}. \tag{2}$$

Even if $S_y$ is only a function of $\zeta(h(\vec{x},t), p(\vec{x}))$, $\Theta$ might be a function of any number of functions $f_i(\vec{x})$ that depend only on the spatial dimensions. The partial derivatives of $\Theta(h(\vec{x},t), f_i(\vec{x})), \forall i \in \mathbb{N}$ are

$$\frac{\partial \Theta}{\partial t} = \frac{\partial \Theta}{\partial h}\frac{\partial h}{\partial t} \tag{3}$$

$$\nabla \Theta = \frac{\partial \Theta}{\partial h}\nabla h + \sum_i \frac{\partial \Theta}{\partial f_i}\nabla f_i \tag{4}$$

Using eqs. (2) and (3) we get

$$S_y(h,p)\frac{\partial h}{\partial t} = \frac{\partial \Theta}{\partial h}\frac{\partial h}{\partial t}, \tag{5}$$

which results in the identification

$$S_y(h,p) = \frac{\partial \Theta}{\partial h}. \tag{6}$$

This preserves the meaning of $\Theta$ as a measure of the amount of water in the peat column. Note that this is made possible because none of the $f_i$ depended on $t$.

Integrating eq.(6),

$$\Theta(h(\vec{x},t), p(\vec{x}), d(\vec{x})) = \int_{d+p}^{h} S_y(h', p)dh'. \tag{7}$$

Notice the dependence of $\Theta$ in $p$ and $d$. Contrary to the version in the manuscript, which only depended on $p$, this sets $\Theta(h = p + d) = 0$. This is necessary because, if we want to interpret $\Theta$ as a measure of the amount of water, we need it to be zero when the water table is at the bottom of the peat column.

Our mistake is now apparent. We ignored the spatial derivatives arising in eq.(4) from the spatially varying $p$ and $d$. The correct expression is

$$\nabla\Theta = \frac{\partial\Theta}{\partial h}\nabla h + \frac{\partial\Theta}{\partial p}\nabla p + \frac{\partial\Theta}{\partial d}\nabla d \tag{8}$$

This issue becomes obvious when transforming the term $\nabla h$ in the groundwater flow equation, as indicated by the reviewer. With eq.(3) and eq.(8), the correct groundwater flow equation in terms of $\Theta$ reads

$$\frac{\partial\Theta}{\partial t} = \nabla\left[\frac{T(\Theta)}{S_y(\Theta)}\nabla\Theta\right] + P - ET - \nabla\left[\frac{T(\Theta)}{S_y(\Theta)}\left(\frac{\partial\Theta}{\partial p}\nabla p + \frac{\partial\Theta}{\partial d}\nabla d\right)\right]. \tag{9}$$

**A.2.2 Specific parameterizations**

The above derivations where independent of any specific parameterization. Here I give all the new, revised expressions, which arise from the modified integration limits of eq.(7). I also write the new terms appearing in eq.(9).

Starting from the parameterization for $S_y$, equation (8) in the original manuscript, eq.(7) results in

$$\Theta(h,p,d) = \begin{cases} h - p + \frac{s1}{s2}(1 - e^{s_2 d}), & h > p \\ \frac{s_1}{s_2}(e^{s_2(h-p)} - e^{s_2 d}), & h \leq p \end{cases} \tag{10}$$

Two things should be noted. First, $\Theta$ is continuous at $h = p$ and 0 when $h$ is at the peat bottom. Second, this expression is different from the one appearing in the manuscript due to the additional $e^{s_2 d}$ arising from the integration limits. As it was explained above, it lets us interpret $\Theta$ as a measure of the peat column water content.

We can now compute the new terms arising in the groundwater flow equation, eq.(9):

$$\frac{\partial\Theta}{\partial p} = \begin{cases} -1 & h > p \\ -s_1 e^{s_2(h-p)} & h \leq p \end{cases} \tag{11}$$

$$\frac{\partial\Theta}{\partial d} = -s_1 e^{s_2 d}, \forall h \tag{12}$$

The new $\Theta$ also influences most of the functions described in Appendix B of the manuscript. We first compute $h(\Theta, p, d)$,

$$h(\Theta,p,d) = \begin{cases} \Theta - \Theta_0 + p, & \Theta > \Theta_0 \\ \frac{1}{s_2}\ln\left(\frac{s_1}{s_2}\Theta + e^{s_2 d}\right) + p & \Theta \leq \Theta_0 \end{cases}, \tag{13}$$

where $\Theta_0 = \frac{s_1}{s_2}\left(1 - e^{s_2 d}\right)$.

With that we can get the peat hydraulic properties $S_y$ and $T$ in terms of $\Theta$. Both $S_y$ and $T$ are the same as in the manuscript under their representation in terms of $\zeta$, but they change slightly in

terms of the newly defined $\Theta$:

$$S_y(\Theta, p, d) = \begin{cases} 1, & \Theta > \Theta_0 \\ s_1 \left( \frac{s_2}{s_1} \Theta + e^{s_2 d} \right), & \Theta \leq \Theta_0 \end{cases},\tag{14}$$

$$T(\Theta, p, d) = \begin{cases} \alpha \left( \Theta - \Theta_0 \right), & \Theta > \Theta_0 \\ t_1 \left( \frac{s_2}{s_1} \Theta + e^{s_2 d} \right)^{t_2/s_2} - t_1 e^{t_2 d}, & \Theta \leq \Theta_0 \end{cases},\tag{15}$$

And, finally, $D(\Theta, d) = T/S_y$. As in the manuscript, $\alpha$ and $\beta$ are fixed by imposing continuity and differentiability of $D(\Theta, d)$ at the domain threshold $\Theta = \Theta_0$.

**A.3  Comparison to the reviewer's approach**

The second reviewer took a different approach to derive the relationship between $\nabla\Theta$ and $\nabla h$, eq.(8). We prefer the derivation given above because it makes all dependences explicit, but in this section we show that the two approaches are equivalent.

The reviewer's derivation began by defining $\Theta$ in terms of $\zeta$ instead of $h$, which is what we presented in eq.(7). Both expressions are related through a simple change of variables: $\zeta(\vec{x}, t) = h(\vec{x}, t) - p(\vec{x})$. With the newly adopted dependence on the peat column thickness $d$, the reviewer's version of eq.(7) reads

$$\Theta(\zeta, d) = \int_d^\zeta S_y(\zeta') d\zeta'.\tag{16}$$

Then, the reviewer invoked the Leibniz integral rule to differentiate under the integral sign. This results in

$$\nabla\Theta = S_y(\zeta)\nabla\zeta - S_y(d)\nabla d.\tag{17}$$

Equation (17) should be equivalent to eq.(8). To enable the comparison, we make the change of variables $\zeta = h - p$ to eq.(17):

$$\nabla\Theta = S_y(h)\nabla h - S_y(h)\nabla p - S_y(d)\nabla d.\tag{18}$$

The condition for the equality of eqs.(18) and (8) requires the following:

$$\frac{\partial\Theta}{\partial h} = S_y(h)$$
$$\frac{\partial\Theta}{\partial p} = -S_y(h)$$
$$\frac{\partial\Theta}{\partial d} = -S_y(d)$$

The first expression is exactly eq.(6). And it can be readily checked, using eqs.(11) and (12), that the other two identities also hold.

**References**

[1] Chris D. Evans et al. "Rates and Spatial Variability of Peat Subsidence in Acacia Plantation and Forest Landscapes in Sumatra, Indonesia". In: *Geoderma* 338 (Mar. 2019), pp. 410–421. ISSN: 00167061. DOI: 10.1016/j.geoderma.2018.12.028.

[2] Yoshiyuki Ishii et al. "Groundwater in Peatland". In: *Tropical Peatland Ecosystem*. Springer Japan, 2016, pp. 265–279.

[3] Julia Jaenicke et al. "Planning Hydrological Restoration of Peatlands in Indonesia to Mitigate Carbon Dioxide Emissions". In: *Mitigation and Adaptation Strategies for Global Change* 15.3 (Mar. 2010), pp. 223–239. ISSN: 1381-2386, 1573-1596. DOI: 10.1007/s11027-010-9214-5.

[4] Ari Laurén et al. "Drainage and Stand Growth Response in Peatland Forests—Description, Testing, and Application of Mechanistic Peatland Simulator SUSI". In: *Forests* 12.3 (Mar. 2021), p. 293. ISSN: 1999-4907. DOI: 10.3390/f12030293.

[5] Ari Laurén et al. "Nutrient Balance as a Tool for Maintaining Yield and Mitigating Environmental Impacts of Acacia Plantation in Drained Tropical Peatland—Description of Plantation Simulator". In: *Forests* 12.3 (Mar. 2021), p. 312. ISSN: 1999-4907. DOI: 10.3390/f12030312.

[6] Susan Page et al. "Anthropogenic Impacts on Lowland Tropical Peatland Biogeochemistry". In: *Nature Reviews Earth & Environment* 3.7 (July 2022), pp. 426–443. ISSN: 2662-138X. DOI: 10.1038/s43017-022-00289-6.

[7] Santosa Sandy Putra, Andy J. Baird, and Joseph Holden. "Modelling the Performance of Bunds and Ditch Dams in the Hydrological Restoration of Tropical Peatlands". In: *Hydrological Processes* 36.1 (Jan. 2022). ISSN: 0885-6087, 1099-1085. DOI: 10.1002/hyp.14470.

[8] Santosa Sandy Putra, Joseph Holden, and Andy J. Baird. "The Effects of Ditch Dams on Water-level Dynamics in Tropical Peatlands". In: *Hydrological Processes* 35.5 (May 2021). ISSN: 0885-6087, 1099-1085. DOI: 10.1002/hyp.14174.

[9] J. P. Skovsgaard and J. K. Vanclay. "Forest Site Productivity: A Review of the Evolution of Dendrometric Concepts for Even-Aged Stands". In: *Forestry* 81.1 (Feb. 2008), pp. 13–31. ISSN: 0015-752X, 1464-3626. DOI: 10.1093/forestry/cpm041.

[10] S Sutikno et al. "Water Management for Hydrological Restoration and Fire Prevention in Tropical Peatland". In: *IOP Conference Series: Materials Science and Engineering* 933.1 (Sept. 2020), p. 012053. ISSN: 1757-8981, 1757-899X. DOI: 10.1088/1757-899X/933/1/012053.

[11] Sigit Sutikno et al. "The Effectiveness of Canal Blocking for Hydrological Restoration in Tropical Peatland". In: *MATEC Web of Conferences* 276 (2019). Ed. by M. Olivia et al., p. 06003. ISSN: 2261-236X. DOI: 10.1051/matecconf/201927606003.

[12] Romuald Szymkiewicz. *Numerical Modeling in Open Channel Hydraulics*. Vol. 83. Water Science and Technology Library. Dordrecht: Springer Netherlands, 2010. ISBN: 978-90-481-3673-5 978-90-481-3674-2. DOI: 10.1007/978-90-481-3674-2.

[13] Iñaki Urzainki et al. "Canal Blocking Optimization in Restoration of Drained Peatlands". In: *Biogeosciences* 17.19 (Oct. 2020), pp. 4769–4784. ISSN: 1726-4189. DOI: 10.5194/bg-17-4769-2020.

---

## Author Response (AR1)

The following is a summary of the main modifications introduced to the manuscript. Our first response to the reviewers contains a point by point answer to all the reviews. Please refer to that document if more details are needed.

**1   Main modifications**

**1.1   Boussinesq equation and numerical schemes**

We corrected the mathematical error spotted by Reviewer 2. We went back to the classical formulation of the Boussinesq equation [1] in terms of $h$ for the peat hydrological module (PHM),

$$S_y(h)\frac{\partial h}{\partial t} = \nabla(T(h)\nabla h) + P - ET. \tag{1}$$

The numerical schemes were modified accordingly, with the difference that the time stepping is explicit now. Explicit time stepping requires tests of accuracy and convergence, which are usually carried out by comparing the results of the simulation with smaller timesteps. As mentioned in the text, we compared the WTD after 5 days of modelling with $\Delta t = 1/1000$ days against the usual timestep ($\Delta t = 1/24$ days), and the relative error was less than 0.1% everywhere. Thus, we accepted the validity of the explicit scheme.

**1.2   Fewer modeled scenarios**

We modelled WTDs for 2 sets of peat hydraulic properties, instead of the previous 4. We think that comparing low and high transmissivities is enough for the aims of the paper, and that having more than two sets of parameters added complexity without generating new insights —and, judging by some of the Reviewers' remarks on the original manuscript, we think they would agree.

**1.3   Other minor changes**

The resolution of the DTM was changed from 50 m x 50 m to 100 m x 100 m. The reason for this is that the original DTM resolution was 100 m x 100 m, and we filled and interpolated it to 50 m x 50 m. Since this preprocessing step is not relevant for most conclusions in the text, it was modified. This is now explained in the manuscript.
Additionally, we added the weather data to the data and code repository, as reminded by Reviewer 1.

**2   New results and discussion**

The results with the corrected version of the Boussinesq equation had many similarities and an important difference. During days of normal water input (i.e., not during the exteme drought of the dry year investigated in the manuscript), the effect of the blocks showed a very similar behaviour to the ones in the original manuscript:

- Dams raised the average WTD in all conditions.

- Heavy rainfalls reduced the difference between the blocked and unblocked scenarios

- high transmissivity peat increased the effect of the blocks, both in the overall average raised WTD, and in the distance at which the rise propagated in the peatland.

- After about 1 km the dams had no effect on WTD
* * *
[1] The reason for this change from the $\Theta$ version to the $h$ version of the Boussinesq equation is the following. The formulation in terms of $\Theta$ was originally introduced for practical reasons: the finite volume library that we used cannot solve equations of the form of Eq.(1) implicitly. By making the change of variables to $\Theta$, we thought we had circumvented the problem. But when we found out about our mistake, it no longer made sense to use this version—the extra terms present in the PDE in terms of $\Theta$ had the same problems than the formulation in terms of $h$. Therefore, we reverted to the classical formulation of the Boussinesq equation, although with an explicit time step.

However, the results did change during the extreme drought of the dry year. While in the original version of the manuscript the drought improved the effect of the dams, this time the results show the opposite: during the long drought the gap between the blocked and unblocked scenarios gets smaller and smaller. The reason for this likely lies in a combination of factors. Our interpretation is that two properties of our model are the main causes [2]. First, our simple model of the canal blocks completely block the water flowing through them. This means that canal segments that lie between two consecutive blocks become completely disconnected from the rest of the network. Therefore, the only water input to this canal segments comes either from precipitation or as inflow from the peatland. In extreme droughts, of course, there is almost no water input from precipitation. This leads us to the second cause: the likely underestimation of water inflow from the peatland to the canals due to the low resolution DTM. The canals are excavated in the peat, which, together with increased peat subsidence close to the canals, means that there is a depression of the peat surface around them. This depression increases the water gradient of the when the CWL is low. However, our low resolution DTM could not resolve those depressions with sufficient detail, and as a result too little water might be entering canals from the peat, especially during dry periods.

**3    Modifications to figures**

Since it is not possible to track figure modifications in a diff document, we briefly explain the differences here.

- Figure 1 was changed according to the suggestion by Reviewer 2. We included diagrams of the DTM resolution and the unstructured mesh.

- Figures 2 and 3 were not modified.

- Figure 4 now contains only two peat hydraulic property parameter sets. The diffusivity was eliminated because it does not exist in the formulation of the Boussinesq equation in terms of $h$, Eq.(1).

- Figure 5 contains the results of the new reality check, together with the separation between modelled and measured data proposed by Reviewer 2.

- Figure 6 now contains the block positions, as suggested by the Reviewers.

- Figures 7, 8 and 9 contain the new results.
* * *
[2]Other phenomena, such as decreased evapotranspiration when the WTD and CWL are low, or an increase in the specific yield, may also be involved in the observed behaviour. However, we don't think they are as critical.

---

## Referee Report (RR1)

Dear Authors,

I am really glad to read the revision of the manuscript. I do believe that this manuscript is substantially ready to be published in *Biogeosciences*.

During the review processes, I am really impressed by the way the author thoroughly discuss their findings and claims, very clear and detail. The authors choose the wordings prudently and reverently when expressing to disagreements.

Meanwhile, please allow me to remark on some points and leave the eagerness to response at the authors' discretion.

Thank you very much.

Regards,

Santosa Sandy Putra

Indonesia

**Referring to document bg-2022-218-author_response-version1.pdf**

**Section 1. Main modifications, 1.1. Boussinesq equation and numerical schemes**

"We went back to the classical formulation of the Boussinesq equation [1] in terms of *h* for the peat hydrological module (PHM)."

> Reviewer 1 remark: To be crystal clear, this statement means that the approach about Θ (presented in section A. Appendix: Mathematical error, in document bg-2022-218-AC1-supplement.pdf), is not being implemented in this paper.

**Section 1. Main modifications, 1.3. Other minor changes**

"Additionally, we added the weather data to the data and code repository, as reminded by Reviewer 1."

> Reviewer 1 remark:  Thank you. I think the presented data have not been accompanied with metadata, including units, benchmark coordinates, operated instruments, and information of data collectors. The metadata will allow reader to interpret the data with minimum helps of the authors.

**Section 3. Modifications to figures**

"Figure 5 contains the results of the new reality check, together with the separation between modelled and measured data proposed by Reviewer 2."

> Reviewer 1 remark: I think the measured and the modelled data are still presented together (no separation) in Figure 5.b.
> I think there are too many lines in Figure 5.b that are probably hard to be distinguished by the readers (the lines are presumably from 141 measurement points). Will it be better to present fewer lines taken from some representative measurement points?
> Will it be better to provide a zoom view that captures a particular shorter time (e.g., 10 or 30 days)?

**Referring to document bg-2022-218-AC1-supplement.pdf**

**Section 1.1 General comments, Subsection 1.1.1**

"The reality check requires that the input data (weather data) and the reference data (WTD) represent the same conditions, and therefore they must be from the same period."

> Reviewer 1 response: Thank you for the definition and also for the improvement presented in the revised manuscript (Line 244 and Line 293).
>
> I think it means that the reality condition can only be approached instead of be captured.
>
> Please consider to substitute "reality check" with either "field check" or "field comparison".

**Section 1.1 General comments, Subsection 1.1.2**

"We think that our claim that "blocks are most effective during dry periods" is well supported by the results we present."

> Reviewer 1 response: Sorry that I cannot provide clear comments earlier about the purpose of seasonal analysis for this study.
>
> Nevertheless, finally the authors presented extreme dry period analysis that is very important related to the results interpretation. The authors make an additional conclusion, which is: "during the long drought the gap between the blocked and unblocked scenarios gets smaller and smaller". It is great to see the fifth outcome discussed in Section 4.2 Block impact on WTD (in Line 391 (new)).

"Nevertheless, we agree that adding a table summarizing the key results will improve the readability of the text. We will compose a table that captures mean WTD and possibly other relevant metrics out of the results presented in Figure 6 and Figure 7."

> Reviewer 1 response: I might be wrong but I have not seen the mentioned table in the manuscript. If the manuscript is too long already, probably the authors can cancel the plan or present the table as Appendixes.

**Section 1.1 General comments, Subsection 1.1.3**

> Reviewer 1 response: Thank you very much for bringing new insights that enhance my perspective.
>
> Please consider to summarize or discuss this paragraph into the manuscript:
>
> "On the other hand, our study area was large and there was a large amount of heterogeneity in the block locations, which capture some of the spectrum of block density present typically in tropical restoration projects: the western part of the catchment, for instance, had more blocks per unit area than the south-eastern part. Indeed, this is exactly what we see in Figure 8: There was a fair amount of variance in the distance up to which blocks affected the WTD."

**Section 1.2 Minor issues, Subsection 1.2.5**

"For a recent review, see [6] and the references therein. Also worth mentioning are the works by some of the authors, [4, 5]."

> Reviewer 1 response: Please allow me to have a different interpretation to reference [4, 5, and 6]. To me, those articles mentioned the possibility of vegetation changes in post drainage conditions. Article 4 explicitly mentioned about *Acacia* plantation. Therefore, I do not think it is suitably just to bring the proposed productivity term into a restoration perspectives. In a restoration perspective, I believe net biomass productivity is more important than gross biomass productivity (omitting biomass degradation).

**Section 1.2 Minor issues, Subsection 1.2.9**

"We will add these coordinates in the final version."

> Reviewer 1 response: Will it possible to mention the coordinates in the text too (e.g., in Line 65 of the latest manuscript)? Not only in the caption of Figure 1?

**Section 1.2 Minor issues, Subsection 1.2.11**

"Reviewer 2 asked to restate the DTM resolution in the Figure caption, a somewhat contradictory requirement to the suggestion by this reviewer."

> Reviewer 1 response: It is great to mention the DTM resolutions in Figure 1. In case there is a comment that is contradictory according to the authors, although so sorry that I personally cannot perceive it, please eliminate it. Thank you.

**Section 1.2 Minor issues, Subsection 1.2.13**

"This is escapes our possibilities. The typesetting strategy in LaTeX is to "float" the figures and tables, i.e., to let the software decide on their best placements."

> Reviewer 1 response: I suggested to put Table 2 in Page 8 of the manuscript (new). It is because Table 2 is firstly referred in Line 158, in Page 8.
>
> The proposed suggestion might help the reader to observe Table 2, which contains peat properties scenarios. I guess the suggested position is somewhat better than locating Table 2 in Page 11, under Section 2.3.1 Weather scenarios.
>
> I extremely agree with the authors. In a research, there are many things out of our control. Visualizing in LaTeX is one of the example. However, it might be helpful to look at these references, as some programmers seem to disagree to our claim of table positioning settings in LaTeX.
>
> https://en.wikibooks.org/wiki/LaTeX/Tables
> https://stackoverflow.com/questions/1673942/latex-table-positioning
> https://tex.stackexchange.com/questions/9485/how-to-fix-table-position

**Section 1.2 Minor issues, Subsection 1.2.18**

Added after L233: "The typical block is made out of surrounding peat, and covers the canal width entirely up to the local peat surface."

> Reviewer 1 response: It will be completely great to describe not only about the 203 permanent peat compaction dams but also about the 87 temporary box dams (see line 68 of the new manuscript). Have those two the same construction structure?

**Section 1.2 Minor issues, Subsection 1.2.19**

> Reviewer 1 response: The current position of Figure 4, under the subsection 2.3.3 Peat hydraulic properties in the new manuscript, is better than my initial suggestion. It looks OK now. I am so sorry initially I forgot that $K$ has different units than $T$.
>
> Moreover, I accepts the authors' explanations about the line styles and does not insist the authors to change the line styles anymore. It might be interesting to find some perspectives in the literature that answer: Why presentations with different line types (e.g. using dashed) might be generally better than presentations with different line colours only? A suggestion is a book by Prof. R. S. Clymo, "Reporting Research: A Biologist's Guide to Articles, Talks, and Posters" (see https://doi.org/10.1017/CBO9781107284234).

**Section 1.2 Minor issues, Subsection 1.2.24**

Reviewer 1 response: I am sorry for the unclear comments.

I just want to highlight the main approach of the authors, which is: "This work does not try to isolate the effect that a single block might have in the WTD of a tropical peatland area, but rather to investigate how the WTD might raise in a typical restoration project under different conditions."

In other words, the authors agree that several canals and/or blocks simultaneously affect the WTD at the study sites.

**Section 1.2 Minor issues, Subsection 1.2.27**

"We forgot to include the precipitation and evapotranspiration data. We thank the Reviewer for the notice. We will upload it."

Reviewer 1 response: Please see the comments of the data repository (in Section 1. Main modifications, 1.3. Other minor changes, referring to document bg-2022-218-author_response-version1.pdf).

It is favourable to make the data being informative to readers, which means low supports from the authors are needed to read the data.

**Referring to document bg-2022-218-manuscript-version3.pdf**
**Minor issues**

1. Line 3: Probably delete the word "however". Please make a double check whether the authors want to write "the WTD monitoring data is …" or "the WTD monitoring data are …".

2. Line 15: Please separate the aerial unit to the $CO_2$ emission unit (e.g., use Mg of $CO_2$ per ha). It is preferably to be applied for the whole documents.

3. Line 70-72: Please give some example of species that are categorized as pioneering native forest species.

4. Line 75: "The peat depth averages at about 5 m." How the authors or the surveyors estimate this value? How the peat depth inventory was conducted? Is it an experts/ practitioners guessing? This is also related to Line 195-196.

5. Line 76: Please mentions the specifications of weather monitoring instruments and WTD loggers concisely. The specifications can be mentioned here or in the metadata file in the shared repository.

6. Figure 1: What is the elevation reference of the DTM? Is that the mean sea level?
   Please consider or rephrase this caption for Figure 1.b:
   Original 100 m × 100 m resolution of the square mesh of the digital terrain model, which was later interpolated to 50 m × 50 m of the triangular mesh of the peat hydrological module (PHM).
   What are the green texts D=0 and D≠0 presented in Figure 1.c? Do you mean T=0 and T≠0?

7. Line 103: Please consider this "where $p$ [m] is the local peat surface (canal bank) elevation above the reference datum."

8. Table 1: For $z$, as far as I am aware, it is common to use "canal depth" over "canal height". As another comparison, river engineers usually use river depth instead of river height.

9. Line 114: Please consider "As an approximation approach, the friction coefficient in this zone must be …"

10. Equation 4: I am still confused with the appearance of comma inside the equation, the comma in between 0 and $h$. Clarify and reformat?

11. Line 169: Please mention that unstructured mesh (triangular unsymmetrical mesh) is used in the peat hydrological module (PHM).

12. Line 194: Please consider "… the raster was interpolated to 50 m × 50 m of the triangular mesh."

13. Line 200: "Each simulation started from the same initial WTD (see Subsection 2.3.5)."

14. Figure 3: In the caption, please use mm d$^{-1}$ or mm$^1$d$^{-1}$ instead of mmd$^{-1}$. It is preferably to be applied for the whole documents.

15. Line 220: I suggest "both" is removed. It is because we usually pair "both" and "and".

16. Line 235-238: Please use m d$^{-1}$ or m$^1$d$^{-1}$ instead of md$^{-1}$. It is preferably to be applied for the whole documents.

17. Line 264: "… the WTD at the dipwell locations for each of the 50 rasters (the 50 daily results) …"

18. Line 288: Please add "The sensors with an annual … patrol post transect." in Figure 5 caption.

19. Line 300-301: Please consider "It is also remarkable that the modelled WTD in most of the peatland area far enough from canals (e.g., > 1km) was …"

20. Line 305: Please consider "…, regardless of peat hydraulic properties or weather conditions (except during the extreme dry periods)."

21. Line 335: Please separate the time interval unit to the $CO_2$ emission unit (e.g., use Mg of $CO_2$ per ha per year). It is preferably to be applied for the whole documents.

22. Figure 9: Please use Mg per ha instead of Mgha$^{-1}$.

23. Line 343-344: Please consider to add an extra comma "Finally, in order to make generalizable claims, the sensitivity to weather conditions and peat hydraulic properties should be accounted."

24. Appendix A: I tried to compare the patterns presented in Equation A5 to the one in Equation A8. Unfortunately, I got confused with the difference in sign (+ and -) and the sign function. Does $f_{sign} = 1$ when $h_i^n < h_k^n$ and $f_{sign} = -1$ when $h_i^n \geq h_k^n$?
Please explain or modify Equation A8. Thank you.

---

## Author Response (AR2)

This document shows our responses to some of the Reviewers' comments.

Some comments that the Reviewers made were accepted, changing the manuscript directly. Those have not been linked here. Instead, we append the manuscript diff file for the last changes at the end of this document. We hope this choice will make the Reviewers' task easier. (Please, ignore the changes in the bibliography in the appended .diff file; those were artificially added by the algorithm producing the diff file. We did not introduce any change to the bibliography.)

**Reviewer 1**

**I think the presented data have not been accompanied with metadata, including units, benchmark coordinates, operated instruments, and information of data collectors. The metadata will allow reader to interpret the data with minimum helps of the authors.**

A brief description of the data has been published in a README.txt file in the code and data repository [4].

**I think the measured and the modelled data are still presented together (no separation) in Figure 5.b. I think there are too many lines in Figure 5.b that are probably hard to be distinguished by the readers (the lines are presumably from 141 measurement points). Will it be better to present fewer lines taken from some representative measurement points? Will it be better to provide a zoom view that captures a particular shorter time (e.g., 10 or 30 days)?**

The Reviewer is right in two observations: 1) Figure 5(b) still plots modelled and measured together; 2) Figure 5(b) contains a lot of information and it is hard to interpret. However, let us note the following:

1) Figure 5(a), as requested by the Reviewers, was a new addition to the revised version of the manuscript. It shows the measured values alone, which, compared to Figure 5(b), improves the general readability of the results.

2) One of the key messages that Figure 5(b) tries to convey is precisely how difficult it is to meaningfully compare the outcome of the modelled and measured WTD. As pointed out in the text, the main reason for this is the fact that the peat surface elevation fluctuates in a similar amount as the WTD does, but over a scale smaller to the resolution of our DTM. Therefore, the best way we found to give some credibility to the modelled results was to check that the overall WTD fluctuation over the simulated year was similar in range and slope to the modelled one. This, of course, does not provide a rigorous statistical validation of the peat hydraulic properties, and we fully acknowledge that—it is one of the reasons why the study was designed to explore different peat hydraulic properties. However, we think that this suggests that the model is not wrong in a very obvious way.

Finally, to answer the Reviewer's suggestion of the changes in the figure more directly: choosing some representative measurement points might end up in a bad match due solely to the mentioned dipwell leveling issue, not to mention the fact that we could be biased to present only the best matching subset of points. Choosing a zoomed view of a smaller timescale would similarly move the focus of the figure away from the most important points that it tries to convey: the difficult comparison between modelled and measured, and the similar general trends that, we think, gives some evidence of the good functioning of the model.

**Thank you for the definition and also for the improvement presented in the revised manuscript (Line 244 and Line 293). I think it means that the reality condition can only be approached instead of be captured. Please consider to substitute "reality check" with either "field check" or "field comparison".**

We prefer the term "reality check" because, in our understanding, it is more straightforward than the alternatives proposed by the Reviewer. The same reason the Reviewer has used to argue against using the word 'reality', i.e., the lack of a solid statistical comparison between measured and modelled values, is the reason why we did not use something like "model validation". We think that the strength implied by the

term "reality" is compensated by the weakness of the term "check". We made no changes to the manuscript over this issue.

**I might be wrong but I have not seen the mentioned table in the manuscript. If the manuscript is too long already, probably the authors can cancel the plan or present the table as Appendixes.**

The Reviewer is right: we forgot to include the table as we had promised. We apologize for that. However, with some more perspective, we now lean towards not including such a table. On the one hand, as the Reviewer mentions, the manuscript is already too long. On the other, adding such a Table would, we think, bring the focus to an oversimplification of the results. The table could give the impression that the average WTD was the key takeaway, while we think that our work is most useful in pointing out general WTD trends under different conditions, which is better shown by the figures.

If the article would have been shorter, a table and an accompanying discussion could have added some useful details. However, with the length of the current version of the manuscript, we worry it might be wrongly interpreted as a summary of the results, and that the benefits of including it would not outweigh the drawbacks. Therefore, we did not add any summarizing table to the manuscript. We would, however, be ready to reconsider our position if persuaded that a table would add value to the manuscript.

**Please consider to summarize or discuss this paragraph into the manuscript: "On the other hand, our study area was large and there was a large amount of heterogeneity in the block locations, which capture some of the spectrum of block density present typically in tropical restoration projects: the western part of the catchment, for instance, had more blocks per unit area than the south-eastern part. Indeed, this is exactly what we see in Figure 8: There was a fair amount of variance in the distance up to which blocks affected the WTD"**

We added a few lines to this effect (see L450 in the new version). The new lines read: "One positive point of this study, however, is that the large study area naturally included some variability of the location of canals and blocks. The western part of the catchment, for instance, had more blocks per unit area than the south-eastern part. This helped to capture some of the spectrum of canal and block densities present typically in tropical restoration projects without having to explicitly include it in the study design. Indeed, this heterogeneity is represented in the variance of the block impact in Figure 8."

**Please allow me to have a different interpretation to reference [4, 5, and 6]. To me, those articles mentioned the possibility of vegetation changes in post drainage conditions. Article 4 explicitly mentioned about Acacia plantation. Therefore, I do not think it is suitably just to bring the proposed productivity term into a restoration perspectives. In a restoration perspective, I believe net biomass productivity is more important than gross biomass productivity (omitting biomass degradation).**

The Reviewer is referring to the following passage in the manuscript (L21): "Canals help to remove water from the naturally waterlogged peat, enhancing site productivity, and opening pathways for wood and crop transportation (Dohong et al., 2017)."

We agree with the Reviewer that in terms of restoration of peatland areas net biomass productivity is much more important. However, that is not the sense in which this passage should be understood. This passage is located in the Introduction, and its aim is to introduce the unfamiliar reader to the reason why canals and ditches are dug in peatlands. By that point in the text, the restoration angle has not yet been introduced. In fact, the very next sentence reads: ´´However, the same mechanisms that make the drainage-based bioproduction economically valuable have severe environmental consequences." Therefore, in L21, site productivity is used in the usual sense of the term.

**I suggested to put Table 2 in Page 8 of the manuscript (new). It is because Table 2 is firstly referred in Line 158, in Page 8.**

The placement of figures and tables and the general look of the manuscript will change once the layout of the final publication is introduced. As far as we know, this step is taken care of by the journal. We therefore have no decision about the position of the table in the text.

**It will be completely great to describe not only about the 203 permanent peat compaction dams but also about the 87 temporary box dams (see line 68 of the new manuscript). Have those two the same construction structure?**

The box dams are wooden frames filled with bags filled with earth or peat [3]. These dams are only a temporary solution to raise the WTL, because they need replacement every 2-3 years. We added some explanation to the text.

**Minor issues**

**Line 3: Probably delete the word "however". Please make a double check whether the authors want to write "the WTD monitoring data is ..." or "the WTD monitoring data are ...".**

We think that the word "however" is fine in L3. There is controversy over whether "data" is singular or plural: https://en.wikipedia.org/wiki/Data_(word)

**Line 15: Please separate the aerial unit to the CO2 emission unit (e.g., use Mg of CO2 per ha). It is preferably to be applied for the whole documents.**

We were also confused about the lack of information about this issue in the Copernicus documentation (https://www.biogeosciences.net/submission.html). Copernicus provides a LaTeX template with a custom-defined "unit" command. Here's the definition of the command:

```
\DeclareRobustCommand*\unit[1]
 {\ensuremath{%
   {\thinmuskip3mu\relax
    \def\mu{\text{\textmu}}\def~{\,}%
    \ifx\f@series\testbx\mathbf{#1}\else\mathrm{#1}\fi}}}
```

If ChatGPT and we understand the code right, this command should take care of the spacing between characters by itself. We therefore think that this is a very valid question for the typesetters of the journal, and we will refer it back to them.

**Line 70-72: Please give some example of species that are categorized as pioneering native forest species.**

From an upcoming publication about the SMPP area vegetation restoration: "Fern species include *Blechnum indicum*, *Nephrolepis biserrata*, *Pteridium aquilinum* and *Stenochlaena palustris*, while sedges are dominated by *Scleria sumatrensis*, with some *Scleria terrestris*. The tall, wild ginger *Alpinia mutica* forms dense stands at a few locations. Main tree species recorded are *Archidendron clypearia*, *Dyera polyphylla*, *Macaranga pruinosa*, *Melaleuca cajuputi*, *Melicope glabra* and *Melicope lunu-ankenda*. These are indigenous peat swamp forest species, and all are pioneer species except swamp *jelutung D. polyphylla* which is also characteristic of mature peat swamp forests."

We included some of these around L70 in the text.

**Line 75: "The peat depth averages at about 5 m." How the authors or the surveyors estimate this value? How the peat depth inventory was conducted? Is it an experts/ practitioners guessing? This is also related to Line 195-196.**

The peat depth was assess using a combination of field sampling and a geographical statistical analysis. The field sampling consisted of 81 plots with three corings at each plot. The regression analysis was conducted using a geographically weighted regression with the R package 'spgwr' [1]. The regression examined relationship between the mean peat depth at each sampling plot and 12 topographic or hydrological variables. Multiple iterations of the geographically weighted regression were performed with different combinations of the 12 predictor variables. We performed model selection in a subsequent assessment of model fit using standard Akaike Information Criterion (AIC) comparisons [2]. The lowest AIC value had the most parsimonious fit, hence a final model that regressed the response variable of peat depth against the predictors: DTM height, slope, SAGA Topographic Wetness Index, and SAGA modified catchment area. Using the regression output and slope equation, a full project area peat depth map was interpolated by first predicting peat depth values at 4709 systematically generated point locations across the entire study area and then by performing an Ordinary- Gaussian Kriging spatial interpolation of peat thickness at a spatial resolution of 66 m.

We did not think that this level of detail was necessary to understand the text. Since this comments are open to the public, we thought it was best to leave these details confined to this discussion.

**Line 76: Please mentions the specifications of weather monitoring instruments and WTD loggers concisely. The specifications can be mentioned here or in the metadata file in the shared repository.**

The weather monitoring was done using simple weather stations, comprised of an ombrometer, a thermometer and a hygrometer. The WTD loggers used were simple perforated PVC pipes.

This was included in the text at around line 80.

**Figure 1: What is the elevation reference of the DTM? Is that the mean sea level? Please consider or rephrase this caption for Figure 1.b: Original 100 m × 100 m resolution of the square mesh of the digital terrain model, which was later interpolated to 50 m × 50 m of the triangular mesh of the peat hydrological module (PHM). What are the green texts $D = 0$ and $D \neq 0$ presented in Figure 1.c? Do you mean $T = 0$ and $T \neq 0$?**

The original elevation reference of the DTM is the sea level. However, absolute elevation plays no role in the hydrological model, and therefore we don't see how mentioning it would improve the text. The green texts are now corrected, as the Reviewer suggests, from D to T. We interpolated the DTM raster from 100 m x 100 m to 50 m x 50 m. These were both rectangular grids. Later, the elevation corresponding to each cell of the triangular mesh of the PHM was sampled from the interpolated 50 m x 50 m DTM raster. We therefore think that the caption of Figure 1 is correct.

**Line 114: Please consider "As an approximation approach, the friction coefficient in this zone must be ..."**

We think that at that point in the text it is clear enough for the reader that our construction of the Manning friction coefficient is an approximation. We prefer not to introduce the proposed change in, as it is already implied by the text.

**Equation 4: I am still confused with the appearance of comma inside the equation, the comma in between 0 and h. Clarify and reformat?**

Thanks for pointing this out. The $\max(a, b)$ function takes two arguments $(a, b)$ and returns the maximum of the two. We added a space after the comma so that the format is more clear.

**Line 194: Please consider "... the raster was interpolated to 50 m × 50 m of the triangular mesh."**

See response above about how the interpolation of the DTM was produced.

**Figure 3: In the caption, please use mm d-1 or mm1d-1 instead of mmd-1. It is preferably to be applied for the whole documents.**

See response above about the unit formatting.

**Line 235-238: Please use m d-1 or m1d-1 instead of md-1. It is preferably to be applied for the whole documents.**

See response above about the unit formatting.

**Line 288: Please add "The sensors with an annual ... patrol post transect." in Figure 5 caption.**

We apologize, but we do not understand what is meant by this comment. The WTD from all sensors, not only those in the patrol post transects are plotted in Figure 5.

**Line 305: Please consider "..., regardless of peat hydraulic properties or weather conditions (except during the extreme dry periods)."**

We disagree with the Reviewer here. The impact of the blocks on the WTD compared to the unblocked case was always positive, for all peat hydraulic properties and all weather conditions. In other words, $\langle \Delta \zeta \rangle$ is always positive in Figure 7 (b, c)

**Line 335: Please separate the time interval unit to the CO2 emission unit (e.g., use Mg of CO2 per ha per year). It is preferably to be applied for the whole documents.**

From the Copernicus manuscript preparation guidelines (https://www.biogeosciences.net/submission.html): "Units must be abbreviated in conjunction with numbers (e.g. the velocity is 10 km h-1) and must be written out without numbers (e.g. the velocity is given in kilometres per hour)".

**Figure 9: Please use Mg per ha instead of Mgha-1.**

See response above about the unit formatting.

**Appendix A: I tried to compare the patterns presented in Equation A5 to the one in Equation A8. Unfortunately, I got confused with the difference in sign (+ and -) and the sign function.**

The Reviewer is right that comparing Eq. A5 with Eq. A8 alone yields the wrong sign in Eq.A8. However, we think that Eq.A8 is correct because according to Eq.(2) the discharge, $Q$, and the gradient of the CWL, $\partial h / \partial x$ have opposite signs. One may not insert Eq.A7 directly into Eq.A5, because Eq.A7 gives the absolute value of the discharge, $|Q|$.

A comment has been added to the text below Eq.A8: "The sign function accounts for the direction of the water flow (note the negative sign of Eq.(2)) ...". Hopefully, this clarifies the issue.

**Reviewer 2**

**Lines 129-130: "The block discharge coefficient, Kb, was determined by adjusting the parameter until the flow rate was acceptable.": How was acceptability determined?**

We forgot to include this change in the previous iteration of corrections. We thank the Reviewer for noticing. The block discharge coefficient, $K_b$, was fixed so that the discharge through a block when the CWL was over

the block head level was comparable to the discharge through any other node of the computational domain under similar slopes—i.e., $Q_b \approx Q$ when $h > p - z_b$. This sentence was added to the text.

**Lines 395-409, Section 4.2, discussion of declining effect of blocks during prolonged droughts: Just a comment, rather than a suggestion: Do I understand correctly that there is no ET in the CNM? At very low water levels, when canal discharge is very slow, could water flow from the canal into the peatland to replace water lost to ET in PHM during the prolonged drought in both scenarios?**

The reviewer is right, the CNM has no ET component. However, mesh cells corresponding to canals may lose water through ET in the PHM step. In principle, the situation described is possible with the model: in the PHM step water can flow between canal and peat mesh cells, in any direction.

[revised manuscript text omitted]